# MolMiner: Towards Controllable, 3D-Aware, Fragment-Based Molecular Design

## Abstract

We introduce MolMiner, a fragment-based, geometry-aware, and order-agnostic autoregressive model for molecular design. MolMiner supports conditional generation of molecules over twelve properties, enabling flexible control across physicochemical and structural targets. Molecules are built via symmetry-aware fragment attachments, with 3D geometry dynamically updated during generation using forcefields. A probabilistic conditioning mechanism allows users to specify any subset of target properties while sampling the rest. MolMiner achieves calibrated conditional generation across most properties and offers competitive unconditional performance. We also propose improved benchmarking methods for both unconditional and conditional generation, including distributional comparisons via Wasserstein distance and calibration plots for property control. To our knowledge, this is the first model to unify dynamic geometry, symmetry handling, order-agnostic fragment-based generation, and high-dimensional multi-property conditioning.

## 1 Introduction

Deep generative models are increasingly central to modern high-throughput screening (HTS) pipelines (Westermayr et al., 2023; Ortega Ochoa et al., 2023), where they generate candidate molecules tailored to specific properties before being filtered through successively more expensive stages: from machine-learning surrogates (Schütt et al., 2017) to quantum chemical calculations such as density functional theory (Kohn & Sham, 1965). These models span a wide range of molecular representations (e.g., SMILES (Weininger, 1988), molecular graphs) and generative approaches (e.g., VAEs (Gómez-Bombarelli et al., 2018; Lim et al., 2018), diffusion models (Hoogeboom et al., 2022b; Wu et al., 2022)).

While many methods address isolated challenges—such as chemical validity, structural diversity, or property control—it remains rare to find models that simultaneously support the full range of capabilities required for practical molecular design. In real-world settings, models must go beyond one-shot generation to support multi-step, interpretable generation processes that flexibly adjust molecular size, incorporate chemically meaningful fragments, and maintain validity throughout. Multi-step generation also enables human-in-the-loop design, offering greater transparency and interactive control. Furthermore, capturing 3D geometry is essential when structure-dependent properties are targeted—yet few autoregressive frameworks incorporate this effectively (Voloboev, 2024).

Another limitation in existing methods is rigid rollout order: most autoregressive models grow molecules from a fixed atom or fragment, reducing flexibility and diversity. We address this with an order-agnostic rollout strategy that allows growth from any starting fragment in any valid order.

Finally, conditional generation is critical for use in HTS pipelines, where desired molecular properties are specified upfront. While most models support only single-target conditioning, we enable multi-property control over twelve physicochemical and structural properties. Users can condition on any subset of properties, and the remaining ones are automatically sampled—facilitating efficient, targeted exploration of chemical space.

We introduce MolMiner, a unified, fragment-based generative model designed for flexible and controllable molecular generation. Our key contributions are:

- **Multi-property conditional generation**: MolMiner supports conditioning on any subset of twelve molecular properties, enabling flexible, user-defined control. It achieves accurate and calibrated generation across a wide range of targets.

- **Symmetry-aware 3D modeling**: We incorporate a dynamic, forcefield-driven geometry update during generation and introduce a standardized protocol to handle fragment symmetries.

- **Order-agnostic generation**: Our rollout strategy avoids fixed atom ordering, improving flexibility and acting as a regularizer.

- **Targeted evaluation protocols**: We propose Wasserstein-based distributional metrics and calibration plots to rigorously assess both unconditional and conditional performance.

## 2 RELATED WORK

Our work builds on fragment-based molecular generation approaches such as JTNN (Jin et al., 2019) and HierVAE (Jin et al., 2020), which assemble molecules sequentially while enforcing chemical validity. Like these models, we use coarse-grained molecular fragments and an autoregressive decoding process. Our model is also order-agnostic, similar in spirit to G-SchNet (Gebauer et al., 2022), allowing flexible rollout without fixing a starting point or strict atom ordering, whereas JTNN and HierVAE are fragment-based but order-fixed, and G-SchNet is order-agnostic but atom-based. Additionally, unlike G-SchNet, we allow the geometry of the partial molecule to remain dynamic during generation, rather than freezing atom positions prematurely. We also explicitly introduce a systematic method to handle fragment symmetries during attachment, an aspect not clearly detailed in earlier fragment-based models such as MoLeR (Maziarz et al., 2024). Finally, we demonstrate conditional generation across twelve molecular properties simultaneously; a scale of multi-target control that, to the best of our knowledge, has not previously been achieved in molecular generative modeling.

## 3 METHOD

We model molecular generation as a fragment-based, order-agnostic (Uria et al., 2014; Hoogeboom et al., 2022a), autoregressive process. Molecules are first decomposed into non-overlapping fragments based on rings and bonds, with attachment points standardized to account for fragment symmetries. Generation proceeds step-by-step: at each step, the model is queried with a focal attachment point on the current partial structure and predicts either a new fragment to attach or a decision to terminate that site. To incorporate 3D information, the partial molecular structure is relaxed using a forcefield (e.g., UFF (Rappe et al., 1992)) and the spatial arrangement is used to inform the prediction. This avoids the rigid, frozen geometries seen in prior methods (Gebauer et al., 2022) and ensures that predictions are conditioned on realistic intermediate structures.

Formally, we define the probability of a molecule $\mathcal{M}$ as the expected likelihood over all valid rollout trajectories $R$, each consisting of a sequence of fragment attachment actions:

$$p(\mathcal{M}) = \mathbb{E}_{R \sim \mathcal{U}(\mathcal{R}(\mathcal{M}))} \left[ \prod_{i=1}^{|R|} p_\theta\big(x_i^{(R)} \big| \mathbf{x}_{<i}^{(R)}, c\big) \right], \tag{1}$$

Here, $x_i = (f_i, a_i)$ is a fragment-attachment pair where $f_i \in \mathcal{V}_f$ is a fragment from the vocabulary and $a_i \in \mathcal{V}_a(f_i)$ is a valid attachment configuration for $f_i$. The model may also select a special "termination" action, which marks an attachment site as closed. The sequence $\mathbf{x}_{<i}^{(R)}$ denotes the partial structure up to step $i$, $c$ represents optional conditioning information (e.g., target properties), $|R|$ is the length of the rollout, and $\mathcal{U}$ denotes uniform distribution over the set of valid rollout orders $\mathcal{R}(\mathcal{M})$. Generation proceeds by alternately attaching fragments or terminating open sites. The process continues until all sites are resolved, yielding a chemically valid, fully assembled molecule. Figure 1 illustrates a step in the autoregressive process of molecular generation with MolMiner.

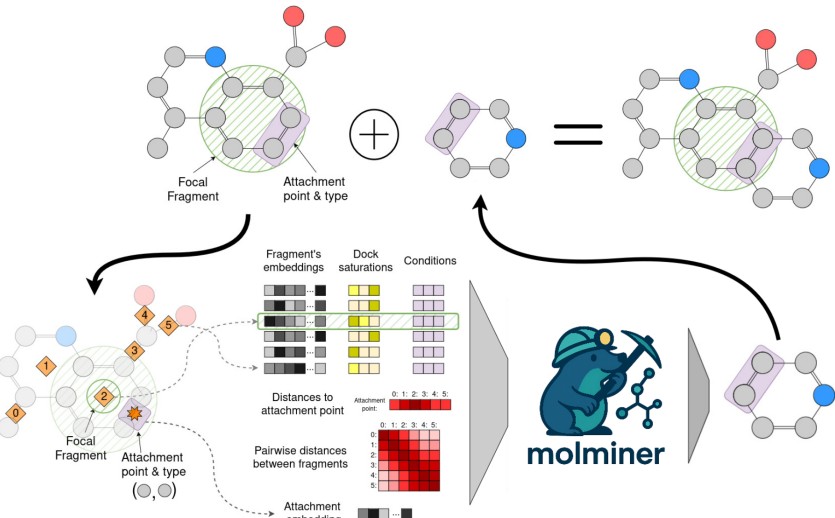

Figure 1: Schematic of MolMiner's fragment-based rollout process. Given a partially grown molecule the model predicts the next attachment. Rollouts proceed in an order-agnostic manner, with growth initiated by an auxiliary predictor that selects the starting fragment.

### 3.1 FRAGMENT-BASED MOLECULAR REPRESENTATION

Molecules naturally exhibit hierarchical structure, often containing repeating substructures such as rings and functional groups. To capture this, we represent molecules as assemblies of chemically meaningful, non-overlapping fragments. Specifically, we apply a coarse-graining procedure that decomposes each molecule into a set of fragments corresponding to rings, identified via the RDKit's Smallest Set of Smallest Rings (SSSR) (Landrum, 2024), and isolated bonds not within a ring. This decomposition strategy is similar to the "small motif" variant explored in HierVAE (Jin et al., 2020), where molecules are fragmented into minimal cyclic and bond-based motifs.

Each extracted fragment is uniquely represented by its Canonical SMILES string (Weininger et al., 1989), computed using RDKit's implementation (Landrum, 2024), providing a compact, human-readable encoding that is invariant to atom indexing within this scheme[1]. However, canonical SMILES do not retain explicit information about how fragments were connected in the original molecule. To preserve attachment information, we track the mapping between each atom's original index in the full molecule and its local index in the extracted fragment. This allows us to recover the attachment points necessary for reassembling molecules from fragments and sets the foundation for our symmetry-aware attachment modeling described next.

We treat each fragment as a discrete token, analogous to tokenization in natural language processing (NLP). This abstraction enables us to associate each fragment with a learnable embedding and formulate molecular generation as a stepwise prediction over a sequence of fragment tokens.

### 3.2 SYMMETRY-AWARE ATTACHMENT MODELING

Although SMILES syntax allows for explicit encoding of atom-specific metadata, such as attachment points using atom-map numbers, incorporating such information would interfere with the canonicalization procedure itself, altering the resulting SMILES string. Furthermore, explicit labeling does not resolve fragment symmetries, where multiple attachment sites may be chemically indistinguishable. A simple example is benzene, where all carbon atoms are symmetry-equivalent.

---

[1]We note that canonicalization procedures differ between cheminformatics toolkits, and different implementations (e.g., RDKit, OpenBabel (O'Boyle et al., 2011)) may produce distinct canonical SMILES for the same molecule (Dashti et al., 2017; Schneider et al., 2015). Throughout this work, all SMILES are generated using RDKit to ensure consistency and reproducibility.

To ensure that fragment attachments are consistently and unambiguously represented, we introduce a symmetry-aware standardization procedure. Since our coarse-graining process extracts fragments corresponding to rings and bonds—both of which are single cycles—the problem of matching atom indices before and after canonicalization reduces to finding valid cyclic permutations. This is because RDKit's canonicalization relies on graph traversal (variants of depth-first or breadth-first search) that follow the topology of the cycle, making reindexing predictable up to a rotation. We exploit this structure by computing similarities between atom environments and identifying cyclic shifts consistent with the fragment's chemical graph.

To reconstruct the atom index correspondences, we compute pairwise similarities between atoms based on their local chemical environments, using Morgan fingerprints (Rogers & Hahn, 2010) and Tanimoto (Tanimoto, 1958) similarity. This yields a similarity matrix that captures correspondences between atom environments in different indexing orders. Valid cyclic permutations are then extracted by identifying rotations that maintain high-similarity mappings across the fragment. Once valid shifts are found, we select a consistent common frame that unifies attachment configurations across symmetric cases, ensuring that generation decisions are invariant to fragment symmetries. Further technical details on the fragment extraction and attachment point handling are provided in Appendix A.6.

### 3.3 Order-agnostic Molecular Rollouts

Molecular generation is framed as a sequence of fragment attachments. At each step, the model is queried with a specific focal attachment point on the current partial structure and predicts either a new fragment to attach or a decision to leave the point vacant. Unlike previous methods that use a fixed rollout order (e.g., breadth-first or depth-first traversal), we adopt an order-agnostic strategy: the next focal attachment point is sampled randomly from the available open sites. The only constraint is that new fragments must attach directly to the existing structure, ensuring the molecule grows as a single connected component. By avoiding any specific traversal scheme and allowing arbitrary selection among open sites, we maximize the flexibility and diversity of possible rollouts for each molecule.

The rollout is initialized by selecting a starting fragment at random from the molecule's fragment set, and identifying its available attachment points. These open sites are placed into an exploration queue. At each step, an attachment point is sampled from the queue, and the model predicts either a fragment to attach or a decision to terminate the site. Unlike linear sequence generation in natural language models, where a single global termination token signals the end of generation, our process is inherently parallel: termination occurs locally at each attachment point. The molecule is considered complete only when all open attachment sites have either been connected to fragments or explicitly closed. This decentralized termination mechanism reflects the graph-like structure of molecules and allows generation to proceed flexibly through multiple concurrent growth paths.

During training, rollouts are precomputed: for each molecule, a sequence of attachment actions and intermediate geometries is generated in advance. This allows efficient learning without the need for force field optimization during training epochs. In contrast, during generation, the molecule is built incrementally, with geometry relaxed after each attachment step via a classical force field. This dynamic procedure ensures that predictions remain geometry-aware throughout autoregressive sampling.

### 3.4 Model Architecture

The model is implemented as a decoder-only transformer (Vaswani et al., 2023) operating over a sequence of fragment tokens. Each fragment is associated with a learnable embedding vector. To incorporate local chemical context, we augment each embedding with three normalized features indicating the fraction of attachment sites that are occupied, free, or sealed. These enriched representations serve as inputs to the transformer layers and help distinguish between fully bonded, partially open, and terminated fragments.

To make the model geometry-aware, we incorporate spatial information directly into the attention mechanism via a global attention bias (Shehzad et al., 2024). Specifically, the attention coefficients

between fragments $i$ and $j$ are given by

$$\alpha_{ij} = \frac{e^{\ g(h_i,h_j)+\theta\cdot D_{ij}}}{\sum_{k=1}^{N} e^{\ g(h_i,h_k)+\theta\cdot D_{ik}}}, \quad D_{ij} = e^{-\frac{||\mathbf{x}_i-\mathbf{x}_j||^2}{2\sigma^2}}, \quad g(h_i,h_j) = \frac{h_i \cdot h_j^\top}{\sqrt{d_h}}, \qquad (2)$$

where $D_{ij}$ is a Gaussian-decayed distance kernel, $\theta$ is a learnable scalar controlling the strength of the geometric bias, and $h_i$ denotes the hidden representation of fragment $i$, as produced by the self-attention mechanism of the previous transformer layer. This mechanism allows the model to attend more strongly to nearby fragments without requiring explicit positional encodings.

Unlike sequence-based tasks in NLP, molecules do not follow a canonical linear order. Instead, spatial relationships emerge from the 3D configuration of fragments. Our attention bias thus acts as a spatial inductive prior that replaces standard positional embeddings with a structure-aware alternative.

During generation, the model is conditioned on the current fragment set, a designated focal fragment, and a specific attachment site (the "hit location"). After processing the structure through the transformer, we perform a focalized readout: the focal embedding attends to all fragments, with attention scores further biased by distances to the hit location. This aggregates global context while emphasizing the local growth site. The resulting vector is concatenated with the conditioning properties, passed through a feed-forward layer, and projected onto the vocabulary of fragment-attachment actions, including the termination action.

### 3.5 TRAINING OBJECTIVE

We train the model to maximize the log-likelihood of each molecule $\mathcal{M}$ under the order-agnostic rollout factorization (Uria et al., 2014; Hoogeboom et al., 2022a), conditioned on target properties $c$:

$$\mathcal{L}(\theta \mid \mathcal{M}) = \log \mathbb{E}_{R\sim\mathcal{U}(\mathcal{R}(\mathcal{M}))} \left[ \prod_{i=1}^{|R|} p_\theta\left(x_i^{(R)} \big| \mathbf{x}_{<i}^{(R)}, c\right) \right] \geq \mathbb{E}_{R\sim\mathcal{U}(\mathcal{R}(\mathcal{M}))} \left[ \sum_{i=1}^{|R|} \log p_\theta\left(x_i^{(R)} \big| \mathbf{x}_{<i}^{(R)}, c\right) \right]$$
$$(3)$$

The expectation is over all valid rollouts $R$ of $\mathcal{M}$, with the lower bound derived via Jensen's inequality (Jensen, 1905). In practice, we use a Monte Carlo approximation of the expectation and randomly sample one rollout per molecule per epoch, providing natural data augmentation by exposing the model to diverse construction orders. At each step, it is trained to predict the next fragment-attachment pair or a termination action, conditioned on the current partial structure and target properties.

To initiate rollouts, we jointly train an auxiliary model to predict a suitable starting fragment from the target properties. This predictor is a feed-forward network that outputs independent probabilities for each fragment in the vocabulary, framing the task as multi-label classification. It is trained with binary cross-entropy loss to encourage high scores for fragments present in the molecule. Both models share the same training splits.

Together, these components enable end-to-end conditional generation—from fragment selection to flexible, geometry-aware rollouts. Importantly, conditioning is implemented in a fully implicit manner: target properties are provided as inputs during training, but no auxiliary loss is applied to enforce property compliance. This allows the model to learn property alignment organically from the data distribution.

### 3.6 SAMPLING PROCEDURE

To generate a molecule conditioned on user-specified properties, we begin by completing the conditioning vector when only a subset of target properties is provided. The missing properties are sampled using a Gaussian Mixture Model (GMM) (McLachlan, 2000) fitted to the empirical distribution of training data. This ensures that completed conditioning vectors remain realistic and consistent with the underlying data distribution. Further details on GMM training and validation are provided in Appendix A.2.

Once the conditioning vector is completed, generation is initialized by selecting a starting fragment. A trained fragment predictor assigns independent probabilities over the fragment vocabulary, from which a seed fragment is sampled. The molecule is then constructed autoregressively.

In Appendix A.7, we evaluate several sampling strategies, including greedy and probabilistic decoding, as well as seed fragment selection from the top-$k$ predictions (with $k = 3, 5, 10$). We also investigate how conditioning values influence the choice of starting fragment in Appendix A.4.

## 4 EXPERIMENTS

We evaluate our model on a subset of the ZINC dataset (Irwin et al., 2012) originally curated for ChemicalVAE (Gómez-Bombarelli et al., 2018), containing approximately 200,000 drug-like molecules. Each molecule is annotated with 12 properties computed using RDKit, which are used both for conditioning and evaluation (see Appendix A.1 for details). We adopt an 80/10/10 train/validation/test split.

### 4.1 TRAINING AND ABLATION SUMMARY

We train an 8-layer decoder-only transformer trained with AdamW (Kingma & Ba, 2017; Loshchilov & Hutter, 2019) and a linear warmup-decay schedule. Hyperparameters were selected via grid search (Appendix A.3), with the final configuration using a 0.3 dropout rate, 64 attention heads, 0.15 warmup ratio and a 5e-5 peak learning rate. Ablation studies confirm three key findings: (i) conditioning on more properties improves performance, consistent with the "tomographic effect" (Ortega-Ochoa et al., 2025), where richer conditioning helps disambiguate structure, (ii) geometry-aware attention aids performance when initialized with positive bias, and (iii) rollout resampling serves as effective regularization, reducing overfitting. These inform our final model, trained with resampling for 50 epochs.

### 4.2 BENCHMARKING UNCONDITIONAL GENERATION

While our model is optimized for conditional generation, we evaluate it under unconditional settings for completeness. We evaluate unconditional generation by measuring how closely the model reproduces the property distributions of the training data. Because direct comparison of molecular graphs is challenging, we instead compare the distributions of twelve physicochemical and structural properties between 5,000 generated molecules and the dataset. These properties include: logP (logarithm of water partition coefficient, used as a measure of lipophilicity) (Wildman & Crippen, 1999), QED (quantitative estimate of drug-likeness) (Bickerton et al., 2012), SAS (synthetic accessibility) (Ertl & Schuffenhauer, 2009), FractionCSP3 ($sp^3$ carbon fraction), molecular weight, TPSA (topological polar surface area) (Ertl et al., 2000), MR (molar refractivity, descriptor accounting for molecular size and polarizability) (Wildman & Crippen, 1999), hydrogen bond donors and acceptors (HBD, HBA), ring count, number of rotatable bonds (flexibility), and number of chiral centers (stereochemical complexity).

To compare distributions, we use the 1D Wasserstein distance for each property, following (Polykovskiy et al., 2018), providing a robust measure of distributional similarity. In addition, we report three standard metrics: Uniqueness (fraction of distinct molecules), Novelty (fraction of valid, unique molecules not present in the dataset), and Diversity (average pairwise Tanimoto distance among generated molecules). We omit validity, as our model enforces valence constraints during generation and consistently produces valid molecules. Molecular identity is determined solely by connectivity, using the first block of the InChIKey (Heller et al., 2015; Pletnev et al., 2012), which encodes the molecular skeleton and excludes variation due to stereoisomerism, tautomerism, and related forms of isomerism.

As our model is inherently conditional, we simulate unconditional generation by sampling conditions to match the training distribution. We evaluate two variants: MolMinerD, which samples conditions directly from the dataset, and MolMinerS, which samples conditions from the GMM.

We benchmark against HierVAE (Jin et al., 2020) an unconditional model, which is the most comparable in terms of generation strategy and architectural design. We exclude MARS (Xie et al., 2021), as it accesses ground-truth molecular properties during generation to guide sampling. Specifically,

MARS evaluates properties such as QED, SA, or activity scores on-the-fly for proposed molecules and uses these values to shape the acceptance probability in a Markov Chain Monte Carlo (MCMC) loop. This fundamentally differs from our approach, in which molecules are generated solely from prompted (i.e., user-specified or sampled) properties, without access to oracle evaluations at inference time. As such, a direct comparison would be misleading in practical scenarios like high-throughput screening, where true property values are unavailable during generation.

We also experimented with MoLeR (Maziarz et al., 2024), using the official implementation and training configuration. The model was run for seven days on an NVIDIA RTX 3090 GPU, completing two 5,000-step validation intervals ("mini-epochs") as defined in the authors' logging protocol. Molecules sampled from the latent prior were often chemically implausible and showed poor alignment with training property distributions. These results are consistent with known limitations of VAE-based molecular models—particularly the mismatch between prior and posterior distributions—and with previously reported decoding issues in MoLeR[2]. We therefore exclude MoLeR from our main quantitative comparisons but include these results in the Appendix A.9.

Additional sampling strategy comparisons for MolMiner are provided in Appendix A.7, along with kernel density plots of the generated property distributions for visual reference.

Table 1: Wasserstein distances between the property distributions of generated molecules (N ≈ 5,000) and the reference dataset are reported, along with uniqueness (%), novelty (%), and mean Tanimoto distance, for HierVAE and Molminer in two different sampling approaches.

| Model | logP | QED | SAS | FractCSP3 | molWt | TPSA | MR | HBD | HBA | #Rings | #RotBonds | #Chiral | %Uniqueness | %Novelty | Diversity |
|---|---|---|---|---|---|---|---|---|---|---|---|---|---|---|---|
| HierVAE | **0.26** | **0.01** | 0.13 | 0.03 | **15** | **2.3** | **3.8** | **0.08** | **0.20** | **0.39** | **0.33** | **0.08** | **100** | **99.9** | 0.88 |
| MolMinerD | 0.31 | **0.01** | **0.07** | **0.02** | 47 | 7.6 | 11.9 | 0.14 | 0.36 | 0.41 | 0.64 | 0.19 | 99 | 99.5 | **0.89** |
| MolMinerS | 0.46 | 0.02 | 0.09 | **0.02** | 65 | 10.9 | 16.3 | 0.16 | 0.56 | 0.59 | 0.88 | 0.26 | 98 | 99.8 | **0.89** |

Our model performs slightly below HierVAE in unconditional generation, with modest differences across most properties. For further analysis we refer to Fig. 15 illustrating the full distributions and a more detailed comparison. The largest gaps—observed in molecular weight, TPSA, and molar refractivity—are partly attributable to approximation error in GMM-based conditioning. While this explains some degradation from MolMinerD to MolMinerS, it does not fully account for the gap. Crucially, MolMiner is optimized for conditional generation, where it enables flexible, multi-property control. We now evaluate its performance in that setting.

## 4.3 BENCHMARKING CONDITIONAL GENERATION

To evaluate conditional generation, we measure how accurately the model produces molecules that match specified target property values. For each of the twelve physicochemical and structural properties, we uniformly sample target values across the range $\mu \pm 2\sigma$, based on their empirical distributions in the dataset. The remaining eleven properties are sampled conditionally from the GMM prior, and the full twelve-dimensional vector is used to guide generation. This process is repeated 30 times per target value, enabling a robust estimate across the full range of each property.

Calibration plots compare the prompted (target) values with the properties predicted from the generated molecules. For continuous properties, we show mean trends with $\pm 1$ standard deviation bands; for discrete properties, we report confusion matrices. This setup evaluates how faithfully the model responds to conditioning across the entire dynamic range of each property, providing insight into its capacity for simultaneous, multi-property control.

As shown in Figure 2, the model achieves calibrated conditional generation for most of the twelve properties. QED is a notable exception, where control accuracy degrades. In addition, molWt and MR exhibit systematic deviations, consistent with their performance in the unconditional benchmark, suggesting areas for further improvement. Overall, to our knowledge, this is the first model to support simultaneous conditioning across as many as twelve molecular properties—representing a significant advance in controllable molecular design.

---

[2]https://github.com/microsoft/molecule-generation/issues/77

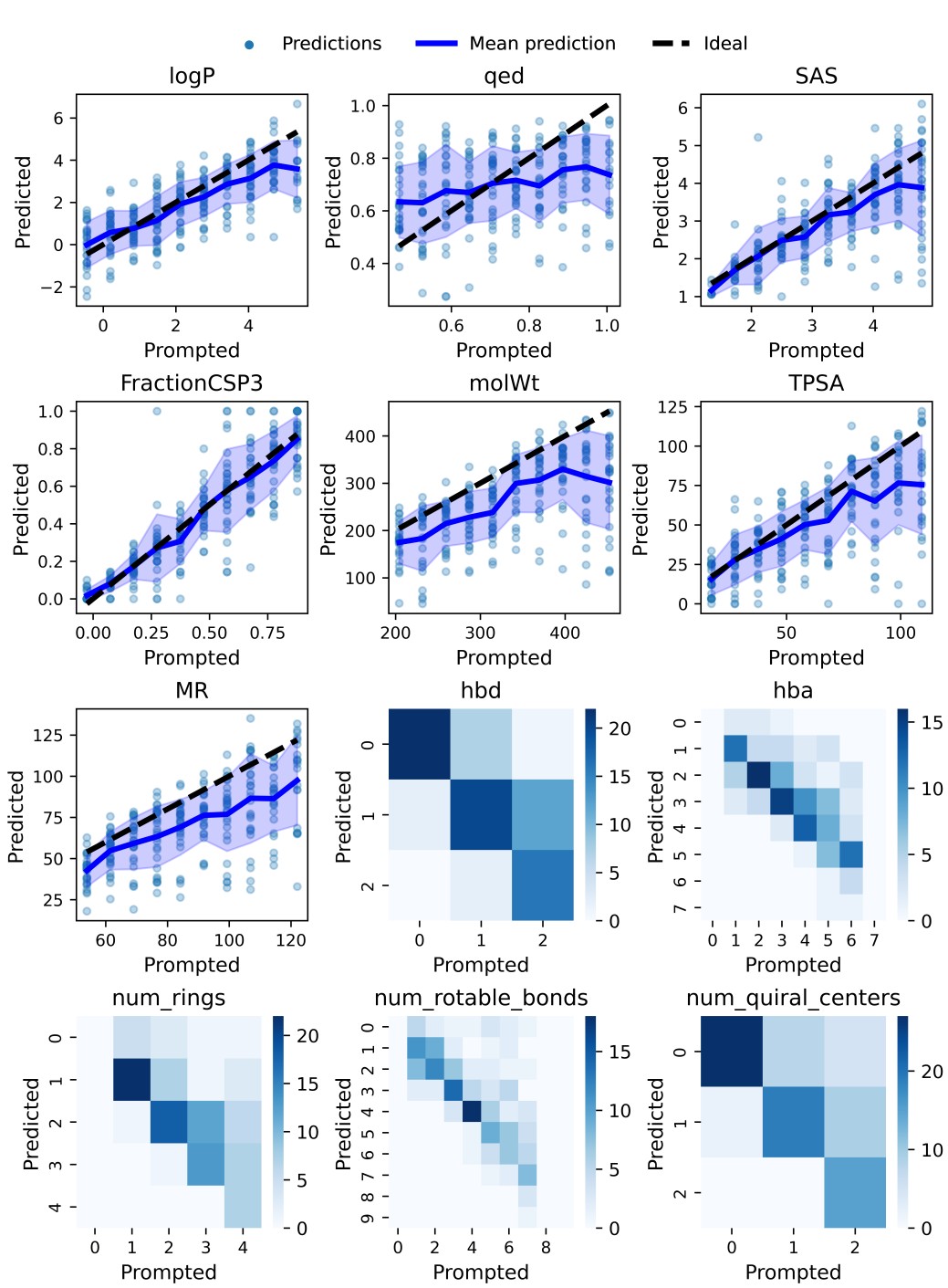

Figure 2: Calibration of predicted molecular properties. Continuous properties show predicted vs. prompted values with mean trends and $\pm 1$ standard deviation bands; discrete properties are summarized as confusion matrices.

## 5    LIMITATIONS

While MolMiner demonstrates strong performance in conditional generation and introduces several architectural innovations, certain limitations remain. Notably, the model underperforms its predecessor in unconditional generation for some properties, particularly molecular weight, MR, and TPSA. We hypothesize that this arises from a tendency to terminate rollouts early, producing slightly smaller molecules on average. This behavior stems from an imbalance in the training data: the order-agnostic rollouts used in MolMiner contain a higher proportion of termination actions than in prior models, potentially biasing the model toward early termination. This effect likely contributes to the systematic deviations observed in the calibration plots, especially for molecular weight. Addressing this may require balancing termination actions during rollout sampling or introducing reinforcement learning based fine-tuning to better calibrate the model's termination policy.

## 6    CONCLUSION

We introduce MolMiner, a novel generative model for inverse molecular design that is autoregressive, fragment-based, geometry-aware, and order agnostic. Crucially, MolMiner supports conditional generation on up to twelve key molecular properties, including logP, QED, SAS, FractionCSP3, molecular weight, TPSA, molar refractivity, hydrogen bond donors and acceptors, ring count, rotatable bonds, and chiral centers.

We show that MolMiner enables controllable and calibrated generation across most of these properties. To make the process more flexible and user-friendly, we introduce a GMM that allows users to specify any subset of properties while the remaining values are sampled conditionally. In unconditional benchmarks, MolMiner performs comparably to existing models across many properties, though some—particularly molecular weight, TPSA, and molar refractivity—still exhibit systematic deviation. More importantly, the model demonstrates strong performance in the more challenging setting of conditional generation.

To our knowledge, this is the first model to unify the following capabilities within a single generative framework: (A) Dynamic incorporation of 3D molecular geometry during autoregressive generation, (B) A symmetry-aware protocol for fragment attachment, (C) Order-agnostic rollout with demonstrated regularization benefits, (D) Scalable, high-dimensional conditional generation using a GMM-based prior. Together, these contributions advance the state of controllable molecular generation and lay the foundation for more interpretable, flexible, and accessible tools for molecular design.

Beyond methodological advances, MolMiner has the potential to accelerate discovery in domains of high environmental and biomedical relevance. By enabling inverse design of molecules with precise control over structural and physicochemical properties, our model could assist in the development of next-generation materials for sustainable energy storage and conversion —such as organic redox flow batteries and organic photovoltaics— facilitate early-stage drug discovery, and support green chemistry initiatives aimed at more environmentally responsible molecular design.

## 7    COMPUTATIONAL REQUIREMENTS

All the models in this work were trained using PyTorch 2.5.0 on a NVIDIA RTX3090. Training these models took approximately 7 days, or 30 epochs, using a batch size of 256 with AdamW as the optimizer, and RAM usage of 70 GB.

## 8    CODE AND DATA AVAILABILITY

All code, model checkpoints, and processed data used in this work are available at https://github.com/xxxx. This includes the ZINC subset with computed properties, dataset splits, training scripts, and evaluation tools needed to reproduce all experiments.

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

## A  APPENDIX

### A.1  CALCULATED PROPERTIES FOR CONTROLLED GENERATION

This work included twelve annotated molecular properties for the compounds in the dataset calculated using RDKit version 2024.3.5, whose statistics for the dataset used in this work are summarized in Table 2.

- **logP**: Logarithm of water partition coefficient, used as a measure of lipophilicity.

- **QED (Quantitative Estimate of Drug-likeness)**: A metric reflecting the drug-likeness of a molecule.

- **SAS (Synthetic Accessibility Score)**: An empirical measure of how difficult a molecule is to synthesize ranging from 1-10. Lower values suggest simpler, more synthesizable structures.

- **FractionCSP3**: The fraction of carbon atoms in $sp^3$ hybridization, used to quantify molecular complexity and three-dimensionality.

- **molWt (Molecular Weight)**: The total weight of a molecule.

- **TPSA (Topological Polar Surface Area)**: The surface area associated with polar atoms, influencing solubility.

- **MR (Molar Refractivity)**: A descriptor related to the molecular volume and polarizability.

- **hbd (Hydrogen Bond Donors)**: The number of hydrogen bond donors.

- **hba (Hydrogen Bond Acceptors)**: The number of hydrogen bond acceptor atoms.

- **num_rings**: The total number of ring structures present in the molecule, which contributes to rigidity.

- **num_rotable_bonds**: The count of rotatable single bonds, a measure of molecular flexibility.

- **num_quiral_centers**: The number of chiral centers in the molecule, indicating stereochemical complexity.

Table 2: Summary statistics in the ZINC dataset used in this work for the twelve molecular properties supported for conditional generation.

| Name | Mean | Std | Min | Max |
|---|---|---|---|---|
| logP | 2.447 | 1.448 | -6.876 | 8.252 |
| qed | 0.736 | 0.135 | 0.112 | 0.948 |
| SAS | 3.071 | 0.864 | 1.133 | 7.289 |
| FractionCSP3 | 0.425 | 0.226 | 0.000 | 1.000 |
| molWt | 328.291 | 62.029 | 150.130 | 499.998 |
| TPSA | 63.173 | 23.033 | 0.000 | 149.700 |
| MR | 87.957 | 17.028 | 17.490 | 151.271 |
| hbd | 1.286 | 0.891 | 0.000 | 6.000 |
| hba | 3.676 | 1.575 | 0.000 | 11.000 |
| num_rings | 2.627 | 0.989 | 0.000 | 9.000 |
| num_rotable_bonds | 4.542 | 1.561 | 0.000 | 11.000 |
| num_quiral_centers | 0.956 | 0.993 | 0.000 | 11.000 |

## A.2 CONDITIONALLY SAMPLING GAUSSIAN MIXTURE MODELS

A trained conditional model supports design control of $d$ properties $\vec{x} = (x_1, x_2, ..., x_d)$. In practice, specifying all $d$ properties is not always possible or convenient. We aim to allow a user to control any subset of controlled properties, which we call 'observed' $\vec{x}_{obs}$, and sample the 'missing' properties $\vec{x}_{miss}$ from the conditional distribution $p(\vec{x}_{miss}|\vec{x}_{obs})$. Then, the fully reconstructed array of properties $\vec{x} = (\vec{x}_{obs}, \vec{x}_{miss})$ can be used as input for the conditional model.

In order to sample all possible combinations of 'missing' properties given any set of possible 'observed' properties it is useful to use a GMM so the conditional distributions are easy to compute. We therefore assume that the $d$ dimensional tuples of control properties for molecules in our dataset are sampled from an unknown distribution that can be approximated by a mixture of finite gaussian distributions with unknown parameters, which are optimized in our case using Expectation Maximization.

$$\vec{x} \sim f(\vec{x}) = \sum_{k=1}^{K} \pi_k \, \mathcal{N}(\vec{x}|\mu_k, \Sigma_k) \tag{4}$$

Then the conditional density:

$$f(\vec{x}_{miss}|\vec{x}_{obs}) = \frac{f(\vec{x}_{obs}, \vec{x}_{miss})}{f(\vec{x}_{obs})} = \sum_{k=1}^{K} \frac{\pi_k \, \mathcal{N}(\vec{x}|\mu_k, \Sigma_k)}{f(\vec{x}_{obs})} =$$

$$= \sum_{k=1}^{K} \frac{\pi_k \, \mathcal{N}(\vec{x}_{obs}|\mu_{k,obs}, \Sigma_{k,obs\,obs}) \, \mathcal{N}(\vec{x}_{miss}|\mu_{k,miss|obs}, \Sigma_{k,miss|obs})}{f(\vec{x}_{obs})}$$

$$= \sum_{k=1}^{K} \frac{\pi_k \, \mathcal{N}(\vec{x}_{obs}|\mu_{k,obs}, \Sigma_{k,obs\,obs}) \, \mathcal{N}(\vec{x}_{miss}|\mu_{k,miss|obs}, \Sigma_{k,miss|obs})}{\sum_{l=1}^{K} \pi_l \, \mathcal{N}(\vec{x}_{obs}|\mu_{l,obs}, \Sigma_{l,obs\,obs})}$$

Where we used:

$$\mathcal{N}(\vec{x}|\mu_k, \Sigma_k) = \mathcal{N}(\vec{x}_{obs}|\mu_{k,obs}, \Sigma_{k,obs\,obs}) \, \mathcal{N}(\vec{x}_{miss}|\mu_{k,miss|obs}, \Sigma_{k,miss|obs})$$

$$f(\vec{x}_{obs}) = \sum_{l=1}^{K} \pi_l \, \mathcal{N}(\vec{x}_{obs}|\mu_{l,obs}, \Sigma_{l,obs\,obs})$$

Re-organizing the conditional density we write:

$$f(\vec{x}_{obs}|\vec{x}_{miss}) = \sum_{k=1}^{K} w_k \, \mathcal{N}(\vec{x}_{miss}|\mu_{k,miss|obs}, \Sigma_{k,miss|obs}) \tag{5}$$

With:

$$w_k = \sum_{k=1}^{K} \frac{\pi_k \, \mathcal{N}(\vec{x}_{obs}|\mu_{k,obs}, \Sigma_{k,obs\,obs})}{\sum_{l=1}^{K} \pi_l \, \mathcal{N}(\vec{x}_{obs}|\mu_{l,obs}, \Sigma_{l,obs\,obs})}$$

$$\mu_{k,miss|obs} = \mu_{k,miss} + \Sigma_{k,miss\,obs}\Sigma_{k,obs\,obs}^{-1}(\vec{x}_{obs} - \mu_{k,obs})$$

$$\Sigma_{k,miss|obs} = \Sigma_{k,miss\,miss} - \Sigma_{k,miss\,obs}\Sigma_{k,obs\,obs}^{-1}\Sigma_{k,obs\,miss}$$

Given $\vec{x}_{obs}$ sampling the conditional distributions can then be done first sampling the k-th gaussian to use, weighted by the scalar $w_k$ and then sampling the corresponding gaussian mixture $\vec{x}_{miss} \sim \mathcal{N}(\mu_{k,miss|obs}, \Sigma_{k,miss|obs})$

### IMPLEMENTATION & HYPERPARAMETERS

We employed the GMM implementation from scikit-learn (version 1.5.2), specifically the *Gaussian-Mixture* class. In order to find the optimal number of components K of the mixture we employ the elbow method with BIC and AIC metrics, as implemented in the aforementioned python package. The GMM models was trained same dataset split train-val-test splits (80-10-10) as the rest of this work. For varying number of mixture components the models were trained on the training set and the BIC and AIC scores noted, as shown in Figure 3.

From Figure 3 we take the K=8 as the best number of components for the GMM in our problem as it is located in the elbow of the plot, indicating an ideal trade-off between complexity and fidelity.

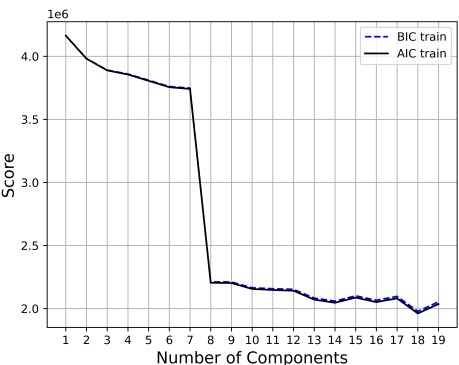

Figure 3: Elbow visualization of the BIC and AIC scores for varying number of GMM components ranging 1-19. Note that at K=8 there is a sharp drop, elbow, which marks an ideal number of components to use for this problem.

FURTHER VALIDATION

Using 8 number of components, we next perform an experiments to evaluate how well does the GMM capture the underlying multi-variate distribution using the validation dataset. We evaluate how good the model is at reconstructing one property given all others. For each property, we mask it and ask the model to reconstruct it, then we produce the quantile-quantile (q-q) plots and compute the Wasserstein (W) distance between the real values and those reconstructed. The result of this first experiment are shown in Figure 4.

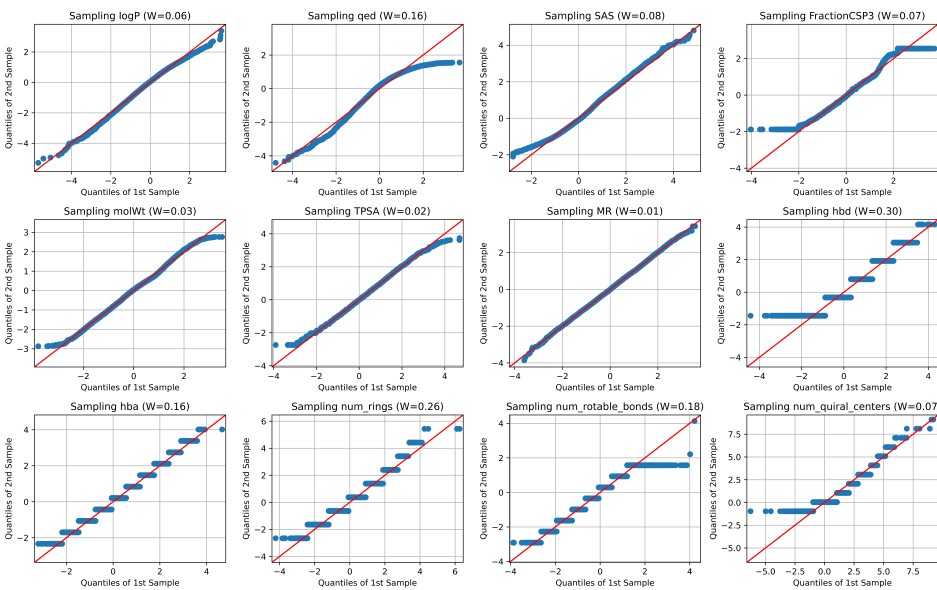

Figure 4: Reconstruction fidelity of 1 missing property given all others. Using the validation dataset, for every property the model is asked to reconstruct one of the properties given the rest, then the reconstructed and real distributions are compared using q-q plots annotated with the Wasserstein (W) distance.

From Figure 4, we can validate that the reconstruction of the missing property using the GMM resembles the true underlying conditional distribution, the quantile-quantile plots fall mostly along the 45 degrees ideal line, and the Wasserstein distances are small, further indicating good agreement.

Note that for properties such as 'number of rings' the q-q plots show horizontal lines as a result of treating the discrete values as resulting from a continuous distribution.

### A.3 ABLATION STUDIES & HYPERPARAMETER SEARCH

We performed a hyperparameter sweep over learning rate (3e-5, 5e-5), dropout (Srivastava et al., 2014)(0.1–0.3), warmup ratio (0.15, 0.2), and number of attention heads (64, 128), selecting 5e-5 peak LR, 0.15 warmup, 0.3 dropout, and 64 heads as the best configuration.

#### A.3.1 THE TOMOGRAPHIC EFFECT

We performed an ablation study to assess the impact of conditioning dimensionality on generative performance. Models were trained conditioning on either 3 or 12 molecular properties, and reconstruction losses were tracked throughout training and validation (Figure 5). Models conditioned on 12 properties consistently outperformed those conditioned on 3, both in terms of lower training and validation reconstruction losses. This performance gap can be attributed to the tomographic effect (as described by (Ortega-Ochoa et al., 2025)), wherein increasing the number of conditioning variables provides the generative model with richer context reducing ambiguity. By offering more informative conditioning, the model achieves improved reconstruction fidelity.

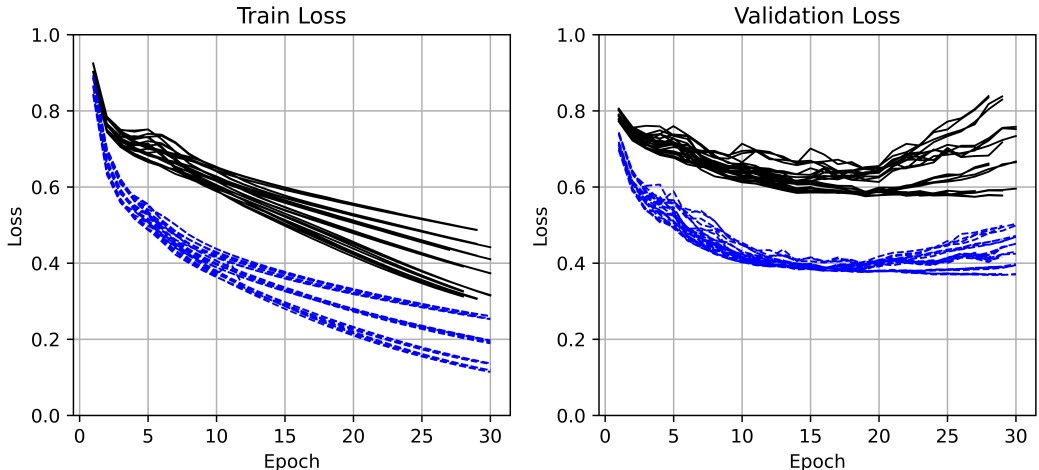

Figure 5: Training and validation curves for models trained with different numbers of conditions. Curves shown in black correspond to models trained with 3 conditions, while those in blue represent models trained with 12 conditions. Models with 12 conditions consistently outperform their 3-condition counterparts in both training and validation, demonstrating improved generalization. This performance gap aligns with expectations from the tomographic effect, where increased conditioning leads to enhanced reconstruction fidelity

#### A.3.2 THE EFFECT OF GEOMETRY

We performed a two-part study to evaluate the influence of geometric information on model performance. First, we conducted an ablation study by comparing models trained with geometry (global geometry attention bias factor initialized at 1 and trained) versus without geometry (factor fixed at 0). As shown in Figure 6, incorporating geometry consistently improved both training and validation reconstruction loss. We then performed a hyperparameter search, initializing the geometry factor across a wide range (from large negative to large positive values) while allowing it to be trained (Figure 7). Models initialized with large negative values—which emphasize distant over proximal atomic relationships—exhibited poor performance or training instability. The best results were obtained with moderate positive initializations, with +1 yielding the strongest performance. These findings confirm the importance of geometric information and demonstrate the sensitivity of model performance to the initialization of geometric priors.

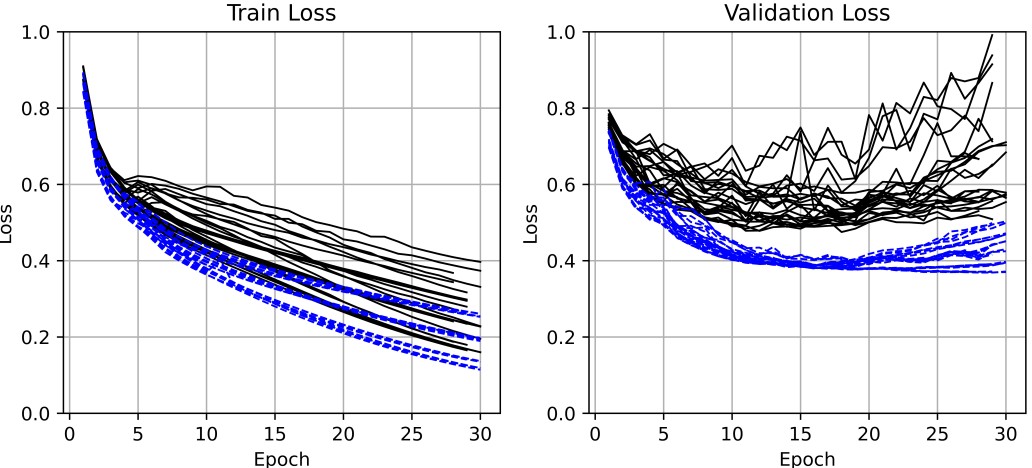

Figure 6: Training and validation curves for models with and without geometric information. Black curves represent models where the geometric weighting factor was fixed at zero and not optimized, effectively removing geometric information. Blue curves correspond to models where this factor was initialized at one and allowed to be learned during training. Models incorporating geometric information consistently outperform those without, highlighting the importance of geometry in model performance. Note the models highlighted in blue are the same as those shown in blue in Figure 6.

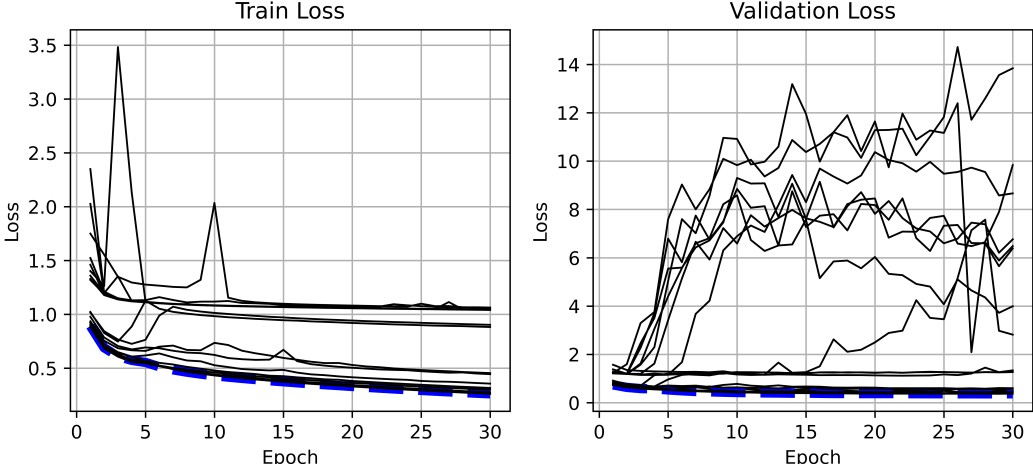

Figure 7: Training and validation curves for models with a trainable geometric weighting factor, initialized with values ranging from ±1000 to 0. All models incorporate geometric information, differing only in the initialization of the geometric factor. The best-performing model, shown in blue, was initialized at +1. All other initializations are shown in black. Notably, models initialized with large negative values diverge during training—consistent with the incorrect weighting of geometry where distant atoms are emphasized more than nearby ones. This highlights both the importance and sensitivity of geometric prior initialization for stable and effective learning.

### A.3.3 THE EFFECT OF RESAMPLING MOLECULAR ROLLOUTS

We evaluated the impact of custom data augmentation by resampling molecular rollouts at each training epoch. As shown in Figure 8, the resampled model outperformed the baseline (no resampling), achieving lower validation loss and improved generalization, as evidenced by a smaller train-validation gap. Notably, the resampled model continued improving beyond 30 epochs, while the baseline's performance plateaued. These results indicate that rollout resampling not only enhances learning but also serves as an effective regularizer, mitigating overfitting.

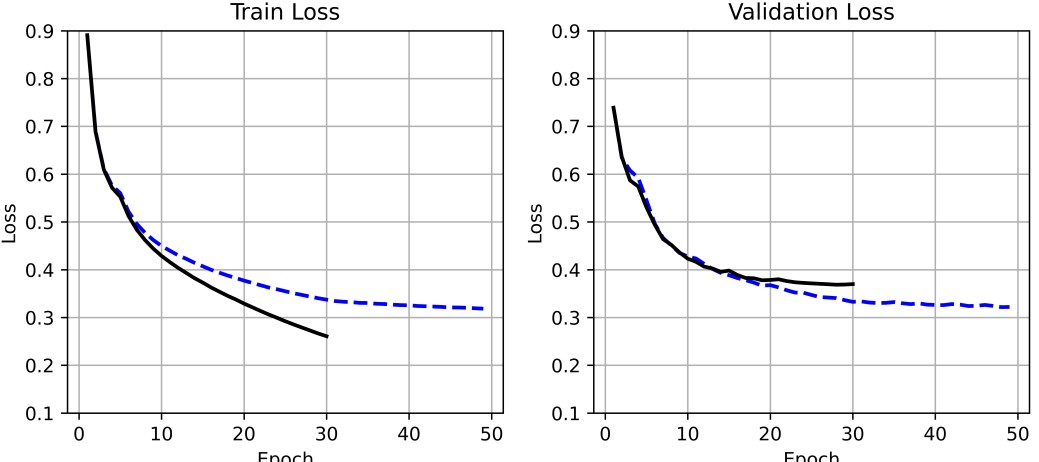

Figure 8: Training and validation curves comparing the best-performing model (black) with a variant that incorporates custom data augmentation (blue), where molecular rollouts are resampled at each epoch. The augmented model outperforms the original both in terms of validation loss and generalization, as evidenced by a significantly reduced train-validation gap. Additionally, the blue curve shows continued improvement beyond 30 epochs, indicating that rollout resampling not only enhances learning but also acts as an effective regularizer. In contrast, the non-augmented model stagnates earlier, as seen in the flattening of its validation curve.

### A.4 EXPERIMENT: INFLUENCE OF CONDITIONING PROPERTIES ON INITIAL FRAGMENT SELECTION

To investigate how individual conditioning properties affect the seed fragment selection process in our molecular generator, we conducted a controlled experiment. The SeedFragNet model, which selects an initial fragment based on a tuple of 12 conditioning properties, was used to generate fragments while systematically varying one property at a time. For each property, we sampled 100 evenly spaced values within $[-2\sigma, +2\sigma]$ and, for each value, generated 100 random samples of the remaining 11 properties using a GMM fitted to the training data. These condition vectors were fed into the SeedFragNet model, and the resulting fragment selections were recorded. The outcome, visualized as flow plots (Figure 9), shows how the frequency of fragment choices evolves as a function of each conditioning property, offering insight into the dependency between the model's conditioning inputs and seed fragment selection behavior.

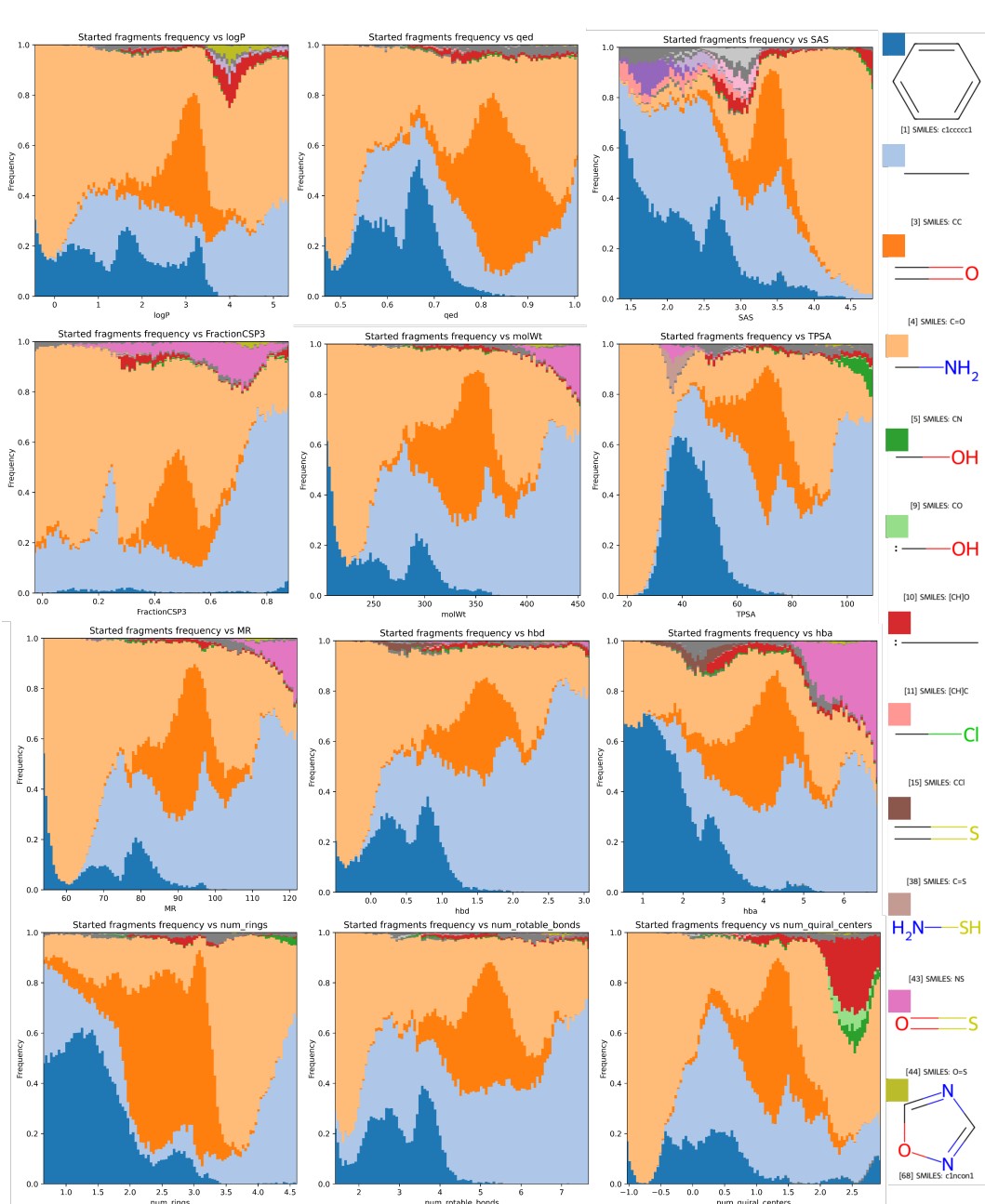

Figure 9: Effect of conditioning properties on fragment selection. Each subplot corresponds to one of the 12 molecular properties used for conditioning. For each property, we vary its value across 100 evenly spaced points within the range $[-2\sigma, +2\sigma]$ ($\sigma$: standard deviation in the dataset), while sampling the remaining 11 properties from a GMM fitted to the training set. For each combination, the fragment-starter model generates a starting fragment, and we repeat this sampling 100 times per property value. The resulting flow plots show the frequency distribution of selected fragments as a function of the conditioned property, revealing how different properties influence the initial fragment choice during generation.

## A.5 Hyperparameter search for SeedFragNet model

We conducted a grid search over the following hyperparameters: dff (number of neurons per layer) [512, 1024, 2058], batch sizes [512, 1024], number of layers [2, 3, 4], dropout probabilities [0.1, 0.3, 0.5], and learning rate factors [0.5, 0.8]. Each model was trained for 100 epochs using an initial learning rate of 1e-4 and a fixed seed of 42, resulting in a total of 108 model configurations. The best-performing model was selected based on validation set performance. Figure 10 presents the training and validation loss curves for all runs, with the final selected model highlighted in blue. This model used dff = 512, batch size = 512, 3 layers, a dropout probability of 0.5 and lr factor 0.5. We used nn.BCEWithLogitsLoss() as the loss function. The training produced a mean training loss of $0.0052 \pm 0.0001$ and a mean validation loss of $0.0047 \pm 0.0001$.

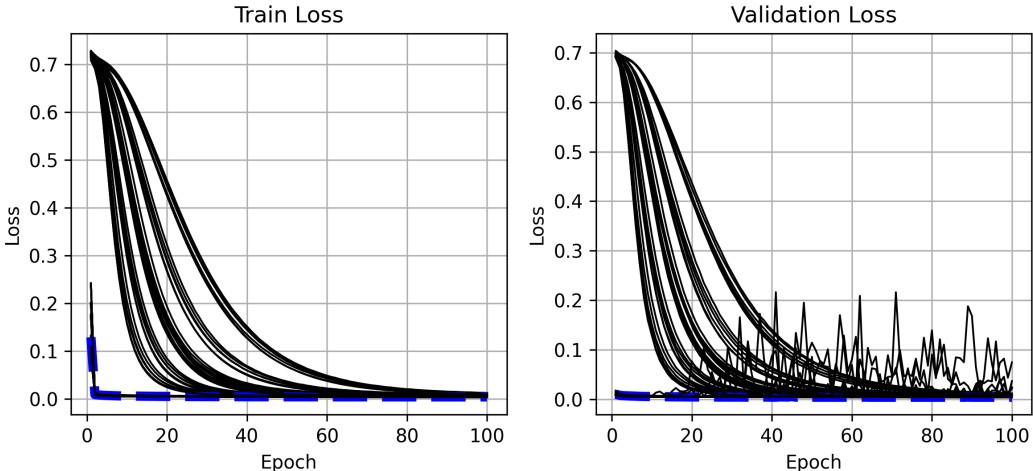

Figure 10: Training and validation loss curves for all hyperparameter configurations. Each line represents a separate model run. The selected final model, chosen based on the lowest validation loss, is highlighted in blue. Notably, several models — including the selected configuration — exhibit substantially lower initial loss values. We hypothesize this is due to favorable random initialization.

## A.6 Fragment decomposition and tokenization of molecular graphs

We represent molecules as assemblies of non-overlapping, interpretable fragments. To construct these, we apply a coarse-graining strategy that systematically decomposes each molecule into a set of irreducible substructures. First, we identify all rings using the SSSR algorithm, which captures the fundamental cyclic motifs of the molecule. Any remaining bonds, connecting atoms not assigned to a ring, are treated as simple two-atom fragments.

This decomposition follows three key principles: First, it ensures **uniqueness**: the same molecule will always be partitioned into the same set of irreducible fragments, independent of atom ordering or indexing. Second, it guarantees **disjointness**, so no two fragments share atoms other than through their attachments, allowing unambiguous reconstruction of the original molecule. Finally, it emphasizes **interpretability**: each fragment corresponds to a recognizable unit, such as a ring or bond, promoting alignment between the model's representation and chemical intuition.

While alternative strategies could use predefined functional groups as fragments, this introduces ambiguity. Functional groups are often composed of smaller substructures that may also occur independently, raising the question of whether to treat the group as a single fragment or to decompose it further. Our approach avoids this ambiguity by consistently decomposing molecules into irreducible fragments.

Additionally, our choice leads to a desirable property: every fragment corresponds to a single cycle—either a ring or a two-atom "bond cycle." This simplifies the identification of symmetries, since

canonicalization and index remapping reduce to cyclic permutations, which are straightforward to compute and resolve.

**Fragment extraction code.** The following function implements the fragment extraction procedure:

```
def _find_clusters(self, molecule) -> list:
    """
    Given an RDKit molecule object, find the list of lists
    containing atom indexes belonging to each grain.
    """
    clusters = [list(x) for x in Chem.rdmolops.GetSSSR(molecule)]
    for bond in molecule.GetBonds():
        a1 = bond.GetBeginAtom().GetIdx()
        a2 = bond.GetEndAtom().GetIdx()
        bond_in_existing_clusters = False
        for cluster in clusters:
            if (a1 in cluster) and (a2 in cluster):
                bond_in_existing_clusters = True
                break
        if not bond_in_existing_clusters:
            clusters.append((a1, a2))
    return clusters
```

Each fragment is represented by its *Canonical SMILES* (RDKit), providing a compact, indexing-invariant encoding that is both human- and machine-readable.

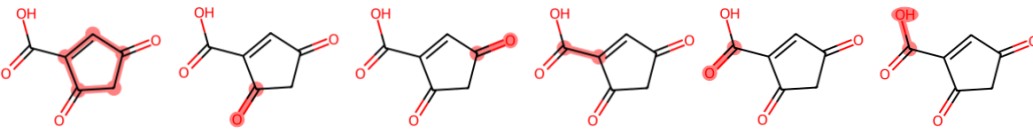

Figure 11: Extracted fragments highlighted on the molecular graph for O=C1CC(=O)C=C1C(=O)O.

TRACKING ATTACHMENT POINTS

While Canonical SMILES provide a convenient representation, they omit explicit attachment information — that is, how fragments connect within the original molecule. To address this, we maintain a mapping between the *global atom indices* (corresponding to the original molecule) and the *local indices* (assigned within each extracted fragment). Atoms shared between fragments naturally define the attachment points.

CANONICALIZATION AND INDEX PERMUTATIONS

When a fragment is re-created from its Canonical SMILES, the local atom indices may change due to the canonicalization process. Fortunately, because every fragment corresponds to a single cycle (either a ring or a two-atom bond cycle), this reindexing is always equivalent to a cyclic permutation of the original atom ordering. To recover the index mapping between the original fragment and its canonicalized form, we compute pairwise atom similarities using the Tanimoto similarity of Morgan fingerprints. This produces a similarity matrix, from which we identify cyclic shifts that yield consistent, high-similarity correspondences. To achieve consistent and symmetry-invariant identification of attachment points across fragments, we construct a map that resolves all permissible index correspondences into a reference form. The procedure is illustrated in Figure 14. Let $\{F_i\}$ denote the set of all valid index mappings between the original fragment atom indices and those assigned upon canonicalization, where $i = 0, 1, 2, \ldots, n$. We select the first mapping, $F_0$, as

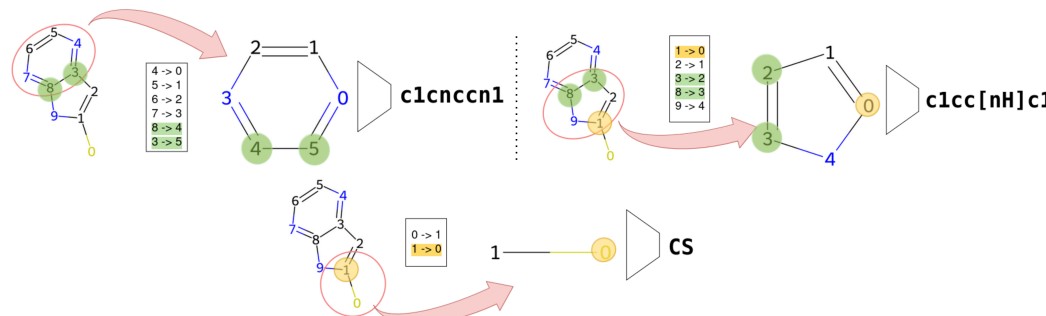

Figure 12: Fragment extraction and attachment point tracking. Atoms shared between fragments define attachment points (highlighted in green).

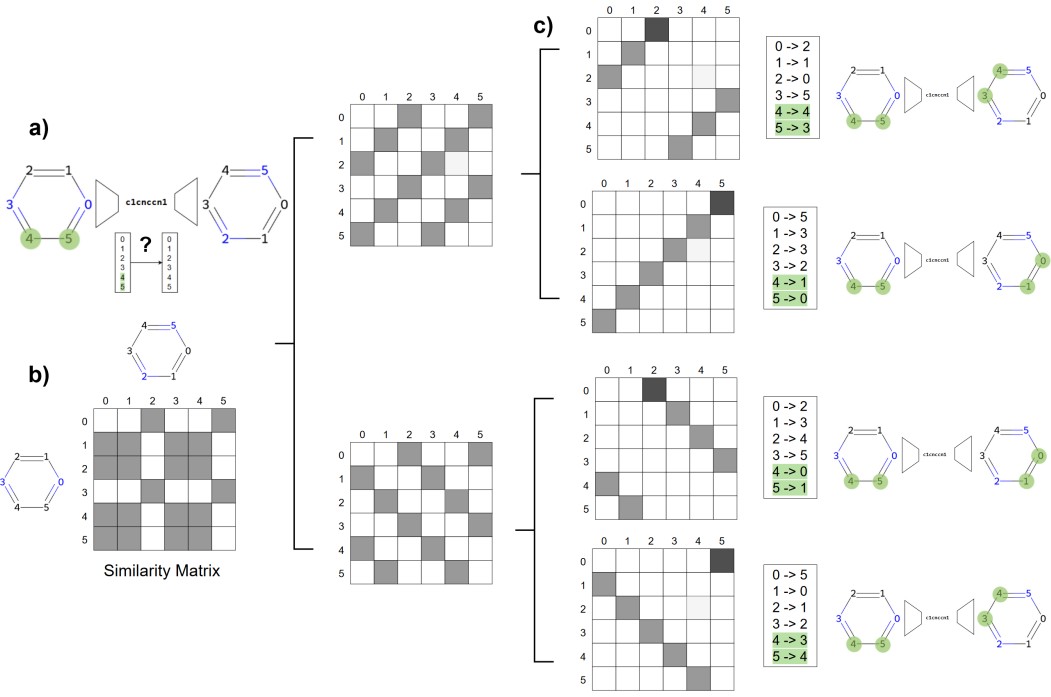

Figure 13: Extraction of all possible maps between local atom indices. (**a**) When a fragment is canonicalized and later reconstructed, its local atom indices may change, resulting in an unknown correspondence between the original and new indices. (**b**) To recover this mapping, we compute the Tanimoto similarity between Morgan fingerprints of every pair of atoms, yielding a similarity matrix. (**c**) Given the prior knowledge that reindexing must correspond to a cyclic permutation, we identify all cyclic shifts that preserve high-similarity correspondences, extracting the valid mappings between the original and new local indices.

the reference and compute its inverse $F_0^{-1}$. For each mapping $F_i$, we then construct a composite mapping defined as:

$$M_i(x) = F_0^{-1}(F_i(x)). \quad (6)$$

This composition aligns all mappings relative to the reference indexing. To derive the final map, for each atom index $x$, we identify the minimal mapped value across all composite mappings:

$$x \longmapsto \min_i (M_i(x)) = \min_i \left( F_0^{-1}(F_i(x)) \right). \quad (7)$$

This selection yields a unique, deterministic correspondence that is invariant to atom reindexing and respects molecular symmetries, including those arising from automorphisms. The resulting standard map provides a robust framework for attachment point tracking, ensuring consistency in fragment reassembly and downstream graph-based computations.

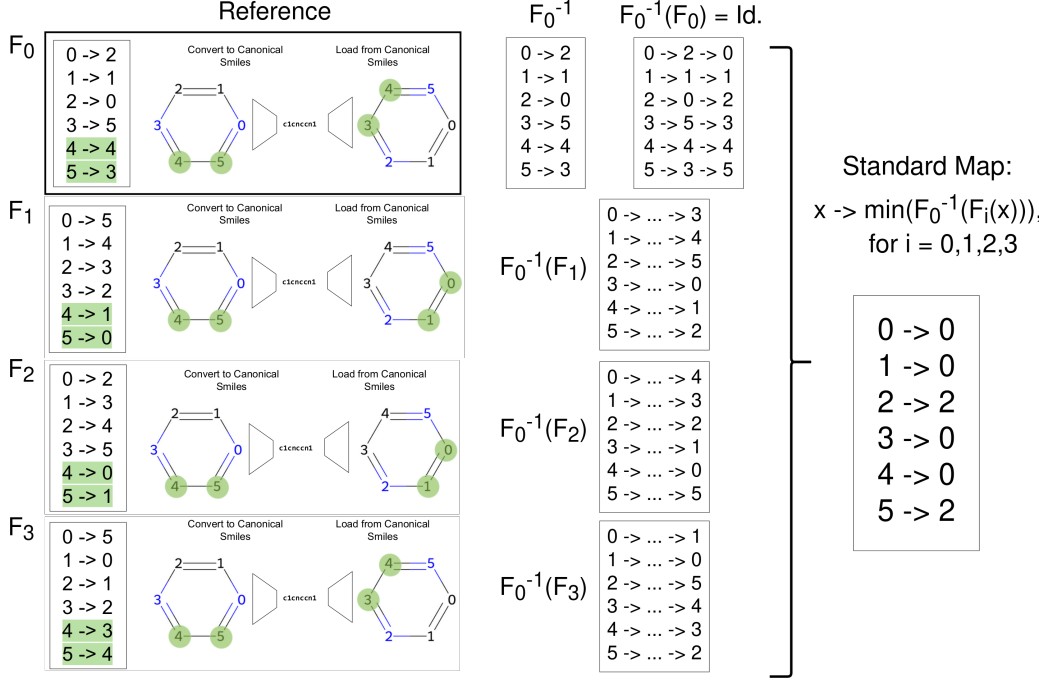

Figure 14: Extraction of fragments and attachment points from a molecular graph, followed by the construction of the standard map to resolve index correspondences in a canonical, symmetry-invariant manner.

## A.7 ABLATION AND SENSITIVITY ANALYSIS OF SAMPLING STRATEGIES

To identify the most effective sampling configuration for our generative model, we conducted a systematic ablation and sensitivity analysis across key components of the sampling process. While the primary goal of this study was to optimize generation quality, the experimental design also provides insight into how individual sampling choices influence the balance between validity, novelty, diversity, and uniqueness.

Our sampling pipeline includes several controllable factors: (i) the source of molecular conditions (either directly from the dataset or sampled from a learned GMM), (ii) the decoding strategy (greedy versus probabilistic sampling during autoregressive generation), (iii) the size of the candidate pool ($k$) from which the initial seed fragment is selected, and (iv) the method for choosing the seed fragment itself (random or weighted by the model's predicted scores). Each of these decisions has the potential to impact the quality and chemical diversity of the generated molecules.

In the following sections, we report the results of varying these factors systematically. The findings not only reveal the most effective configuration for our task but also clarify how different sampling strategies contribute to performance trade-offs. This analysis serves both as a guide for practical deployment and as an ablation study highlighting the robustness and flexibility of our model.

Table 3: Wasserstein distances between the property distributions of generated molecules (N ≈ 5,000) and the reference dataset are reported, along with uniqueness (%), novelty (%), and mean Tanimoto similarity, across various sampling strategies. For example, S/3/NW/NG indicates that conditions were sampled from the GMM (S) rather than the dataset (D); the initial fragment was selected from the top $k = 3$ candidates; selection among these was random and unweighted (NW); and subsequent fragments were sampled in a non-greedy (NG), probability-weighted manner during the autoregressive process. For each metric, the best-performing configuration is highlighted in bold. Overall, the best configuration was D/5/W/NG, while among sampled-only settings, S/10/W/NG performed best.

| Config. | logP | QED | SAS | FractCSP3 | molWt | TPSA | MR | HBD | HBA | #Rings | #RotBonds | #Chiral | %Uniqueness | %Novelty | Diversity |
|---|---|---|---|---|---|---|---|---|---|---|---|---|---|---|---|
| S/3/NW/NG | 0.53 | 0.02 | 0.08 | **0.02** | 67.23 | 10.99 | 16.99 | 0.15 | 0.59 | 0.61 | 0.92 | 0.25 | 98.0 | 99.8 | **0.89** |
| S/3/NW/G | 0.60 | 0.03 | 0.26 | 0.03 | 83.88 | 15.28 | 20.46 | 0.20 | 0.92 | 0.78 | 0.96 | 0.33 | 91.6 | 99.8 | 0.88 |
| D/3/NW/NG | 0.33 | **0.01** | **0.06** | **0.02** | 49.79 | 8.02 | 12.60 | **0.14** | 0.41 | 0.42 | 0.73 | 0.20 | 98.7 | 99.3 | 0.88 |
| D/3/NW/G | 0.44 | **0.01** | 0.20 | 0.04 | 63.35 | 10.51 | 15.53 | 0.17 | 0.62 | 0.52 | 0.86 | 0.24 | 92.1 | 98.3 | 0.87 |
| S/3/W/NG | 0.51 | 0.02 | 0.08 | **0.02** | 67.48 | 11.75 | 17.03 | 0.16 | 0.60 | 0.59 | 0.92 | 0.26 | 97.3 | 99.9 | **0.89** |
| S/3/W/G | 0.62 | 0.03 | 0.28 | 0.03 | 84.94 | 15.38 | 20.61 | 0.21 | 0.91 | 0.80 | 0.96 | 0.34 | 90.7 | 99.8 | 0.88 |
| D/3/W/NG | 0.34 | **0.01** | 0.07 | **0.02** | 49.00 | 7.76 | 12.39 | **0.14** | **0.36** | 0.41 | 0.75 | 0.21 | 98.8 | 99.7 | **0.89** |
| D/3/W/G | 0.50 | 0.02 | 0.17 | 0.03 | 66.57 | 11.22 | 16.45 | 0.17 | 0.66 | 0.56 | 0.91 | 0.26 | 90.9 | 98.4 | 0.87 |
| S/5/NW/NG | 0.54 | 0.02 | 0.09 | **0.02** | 66.45 | 10.87 | 16.89 | 0.16 | 0.57 | 0.61 | 0.90 | 0.24 | 97.9 | 99.9 | **0.89** |
| S/5/NW/G | 0.63 | 0.04 | 0.24 | 0.03 | 85.85 | 15.88 | 21.20 | 0.23 | 0.94 | 0.80 | 1.03 | 0.32 | 90.7 | 99.9 | 0.88 |
| D/5/NW/NG | 0.37 | **0.01** | 0.07 | **0.02** | 49.64 | 7.75 | 12.72 | 0.16 | 0.37 | 0.42 | 0.76 | 0.20 | 98.6 | 99.4 | **0.89** |
| D/5/NW/G | 0.44 | 0.02 | 0.18 | 0.04 | 65.50 | 11.47 | 16.19 | 0.18 | 0.65 | 0.55 | 0.87 | 0.23 | 91.3 | 98.5 | 0.87 |
| S/5/W/NG | 0.51 | 0.02 | 0.09 | **0.02** | 68.08 | 11.60 | 17.13 | 0.16 | 0.59 | 0.61 | 0.92 | 0.27 | 97.4 | 100.0 | **0.89** |
| S/5/W/G | 0.61 | 0.03 | 0.25 | 0.03 | 85.22 | 15.60 | 20.62 | 0.21 | 0.92 | 0.79 | 0.98 | 0.33 | 90.4 | 99.6 | 0.88 |
| **D/5/W/NG** | **0.31** | **0.01** | 0.07 | **0.02** | **47.23** | 7.59 | **11.91** | **0.14** | **0.36** | 0.41 | **0.64** | 0.19 | 98.9 | 99.5 | **0.89** |
| D/5/W/G | 0.47 | 0.02 | 0.19 | 0.03 | 67.27 | 11.95 | 16.66 | 0.21 | 0.68 | 0.55 | 0.97 | 0.26 | 91.0 | 98.5 | 0.87 |
| S/10/NW/NG | 0.58 | 0.02 | 0.10 | **0.02** | 68.67 | 10.77 | 17.70 | 0.16 | 0.55 | 0.62 | 0.96 | 0.25 | 98.2 | 99.8 | **0.89** |
| S/10/NW/G | 0.65 | 0.04 | 0.29 | 0.05 | 89.62 | 16.44 | 22.60 | 0.27 | 0.95 | 0.82 | 1.16 | 0.34 | 89.1 | 99.6 | 0.88 |
| D/10/NW/NG | 0.41 | **0.01** | 0.07 | 0.03 | 51.93 | **7.53** | 13.57 | 0.15 | **0.36** | 0.45 | 0.78 | **0.18** | 98.7 | 99.6 | **0.89** |
| D/10/NW/G | 0.47 | 0.02 | 0.24 | 0.05 | 68.32 | 11.85 | 17.44 | 0.23 | 0.66 | 0.58 | 0.97 | 0.27 | 90.6 | 98.9 | 0.88 |
| **S/10/W/NG** | 0.46 | 0.02 | 0.09 | **0.02** | 64.67 | 10.88 | 16.27 | 0.16 | 0.56 | 0.59 | 0.88 | 0.26 | 97.6 | 99.8 | **0.89** |
| S/10/W/G | 0.63 | 0.03 | 0.25 | 0.03 | 84.00 | 14.84 | 20.40 | 0.21 | 0.87 | 0.78 | 0.96 | 0.33 | 90.7 | 99.7 | 0.88 |
| D/10/W/NG | 0.33 | **0.01** | **0.06** | **0.02** | 48.55 | 7.87 | 12.24 | 0.15 | **0.36** | 0.40 | 0.72 | 0.19 | 98.7 | 99.5 | **0.89** |
| D/10/W/G | 0.47 | 0.02 | 0.19 | 0.03 | 64.86 | 10.88 | 16.02 | 0.18 | 0.63 | 0.55 | 0.89 | 0.26 | 91.4 | 98.6 | 0.87 |

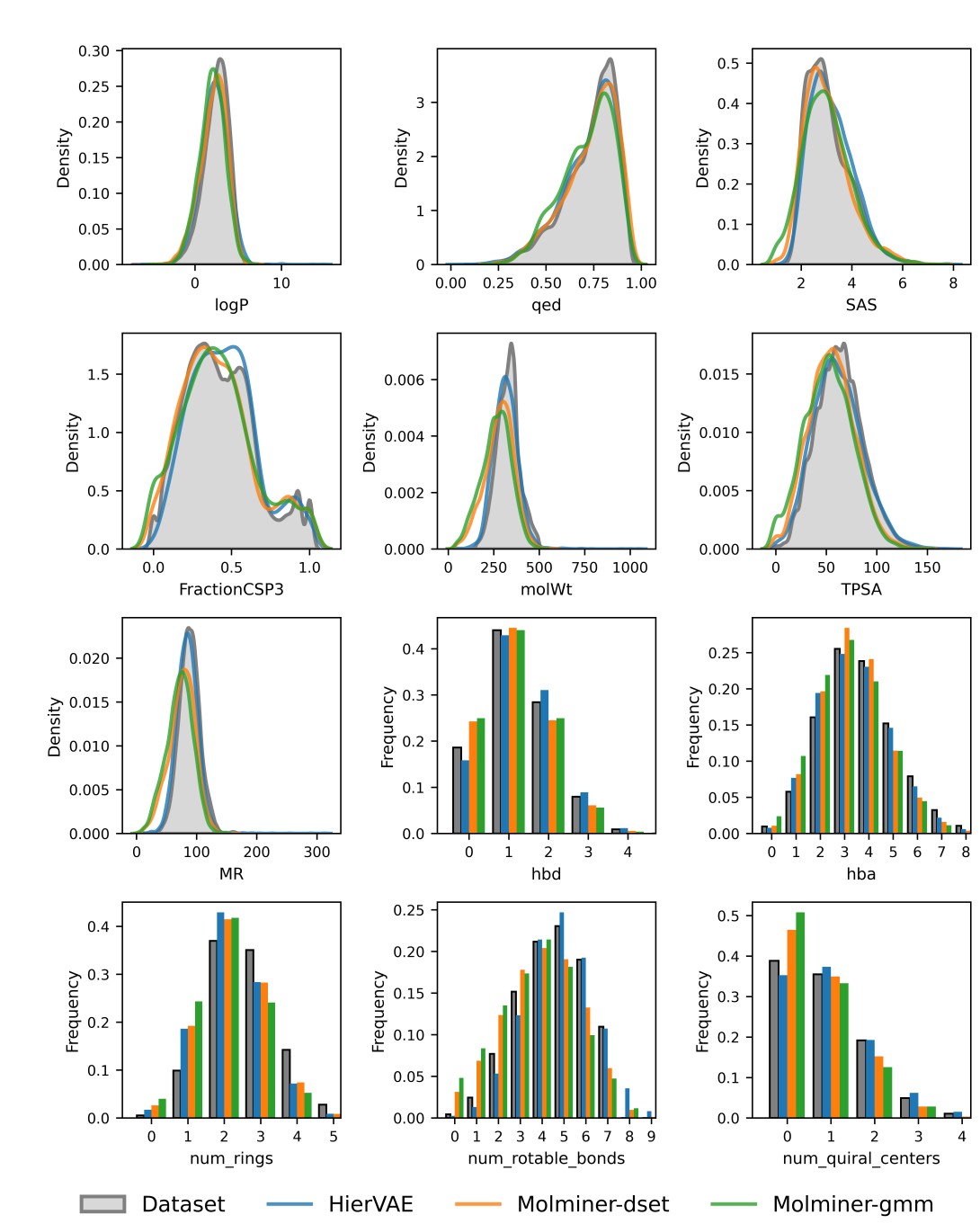

Figure 15: Kernel density estimates (KDE) and histograms of the distributions for twelve molecular properties across 5,000 generated molecules for each model and sampling strategy. We compare the dataset (reference) distribution to HierVAE, MolMinerD, and MolMinerS—the best-performing configurations of our model under dataset-sampled (D) and GMM-sampled (S) conditions, as identified in Table 3. These plots offer a visual complement to the 1D Wasserstein distances reported in the main text, highlighting how well each method replicates the property distributions of the dataset in the unconditional generation setting.

A.8  GALLERY OF GENERATED MOLECULES

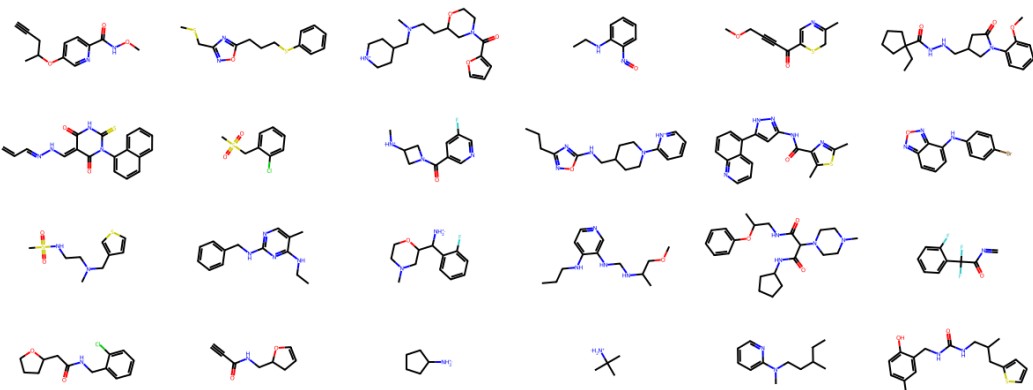

Figure 16: Example of 24 generated molecules by MolMiner in the unconditional generation experiment.

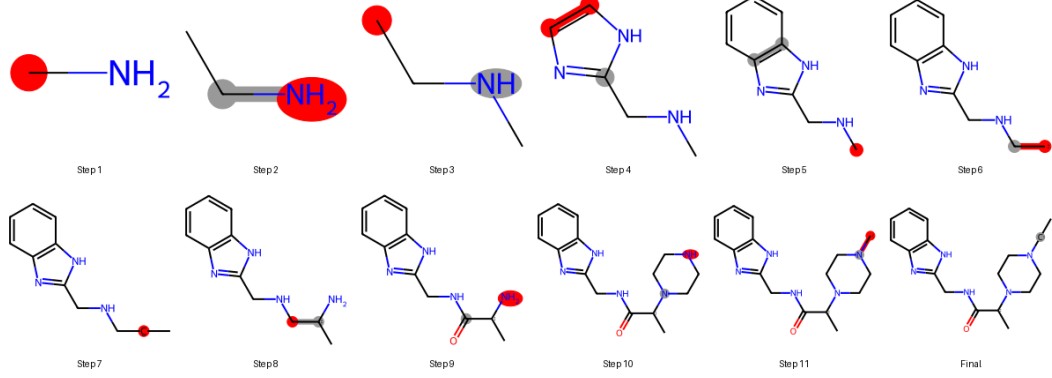

Figure 17: Example of autoregressive generation steps by MolMiner. Note that termination steps are omitted for the visualization. Atoms highlighted in red denote the focal point at which the molecule is grown.

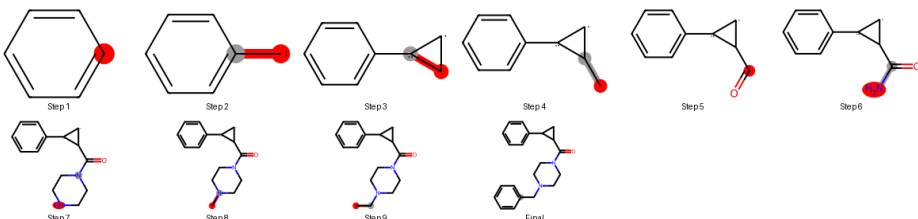

Figure 18: Example of autoregressive generation steps by MolMiner. Note that termination steps are omitted for the visualization. Atoms highlighted in red denote the focal point at which the molecule is grown.

## A.9 UNCONDITIONAL GENERATION EXPERIMENT WITH MOLER

Table 4: Wasserstein distances between the property distributions of generated molecules (N ≈ 5,000) and the reference dataset are reported, along with uniqueness (%), novelty (%), and mean Tanimoto distance, for HierVAE and Molminer in two different sampling approaches and MoLeR

| Model | logP | QED | SAS | FractCSP3 | molWt | TPSA | MR | HBD | HBA | #Rings | #RotBonds | #Chiral | %Uniqueness | %Novelty | Diversity |
|---|---|---|---|---|---|---|---|---|---|---|---|---|---|---|---|
| HierVAE | **0.26** | **0.01** | 0.13 | 0.03 | **15** | **2.3** | **3.8** | **0.08** | **0.20** | **0.39** | **0.33** | **0.08** | **100** | 99.9 | 0.88 |
| MolMinerD | 0.31 | **0.01** | **0.07** | **0.02** | 47 | 7.6 | 11.9 | 0.14 | 0.36 | 0.41 | 0.64 | 0.19 | 99 | 99.5 | **0.89** |
| MolMinerS | 0.46 | 0.02 | 0.09 | **0.02** | 65 | 10.9 | 16.3 | 0.16 | 0.56 | 0.59 | 0.88 | 0.26 | 98 | 99.8 | **0.89** |
| MoLeR | 4.00 | 0.58 | 5.67 | 0.42 | 1303 | 131.1 | 383.2 | 11.65 | 15.90 | 30.07 | 4.46 | 9.02 | 96.5 | **100.0** | 0.80 |

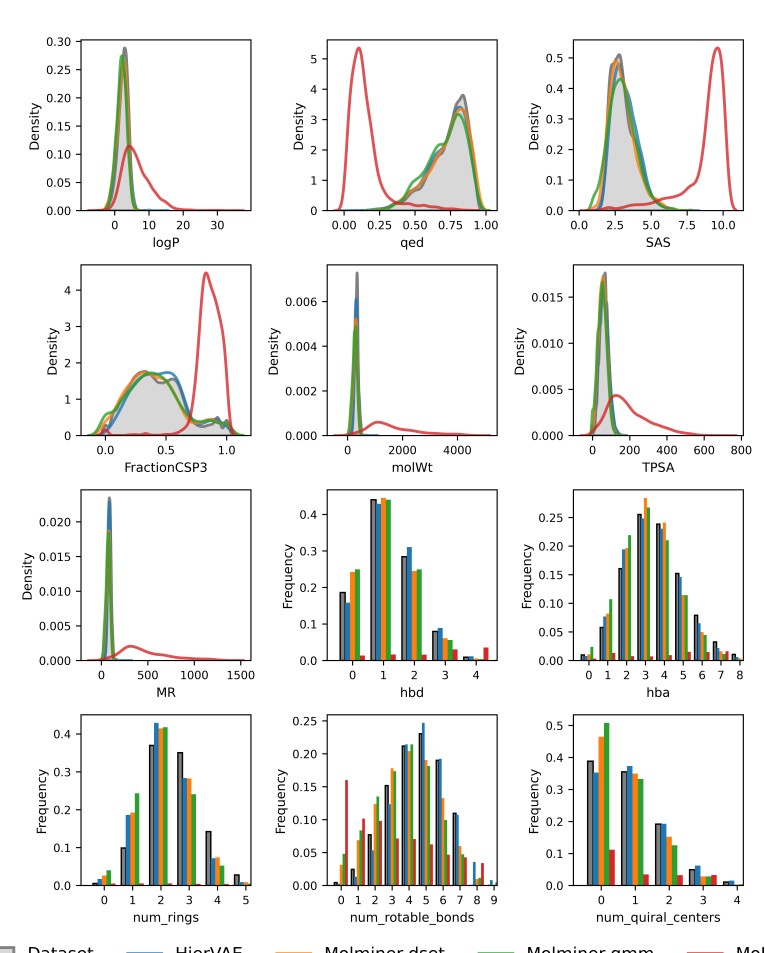

Figure 19: KDE and histograms of the distributions for twelve molecular properties across 5,000 generated molecules for each model and sampling strategy. We compare the dataset (reference) distribution to HierVAE, MoLeR, MolMinerD, and MolMinerS—the best-performing configurations of our model under dataset-sampled (D) and GMM-sampled (S) conditions.

