# OpenReview forum: "MolMiner: Towards Controllable, 3D-Aware, Fragment-Based Molecular Design"
_ICLR.cc/2026/Conference — ICLR 2026 Conference Withdrawn Submission_

### Official Review · Reviewer_Y23o · 2025-10-16

**Soundness:** 2
**Presentation:** 2
**Contribution:** 2
**Rating:** 2
**Confidence:** 4

**Summary:**

The paper introduces a novel autoregressive generative model for inverse molecular design tasks, which is order-agnostic and supports conditioning on up to twelve molecular properties. The authors show the effectiveness of their approach by evaluating on unconditional and conditional generative tasks.

**Strengths:**

- The paper is well-written and well-structured.
- The authors provide a novel order-agnostic, geometry-aware model for molecular generation that supports multiple molecular properties conditioning.
- The authors show the proposed method’s versatility across 12 molecular properties.

**Weaknesses:**

- While the authors introduce several new architectural design choices, the paper does not present substantial methodological novelty. The proposed model largely builds upon MoLeR, which also performs autoregressive (AR) generation by sequentially adding fragments to active sites. Moreover, the idea of sampling missing properties using a Gaussian Mixture Model (GMM) was already explored in MoLeR.

- The authors claim that their method enables flexible structural target conditioning, yet the evaluation is limited to conditioning on chemical properties.

- The related work section omits several important prior works. This includes sequential molecular generation methods such as GraphAF [1] and MiCaM [2], as well as one-shot molecular graph generation models like MAGNET [3] and PSVAE [4].

- The unconditional generation experiment (Table 1) is missing many key baselines, including the aforementioned works [1-4]. Furthermore, across most evaluation metrics, HierVAE outperforms the proposed approach.

- The authors claim that their model is the first to support multi-property conditioning. However, this capability has already been demonstrated in diffusion-based molecular generation methods: https://openreview.net/pdf?id=X41c4uB4k0

- The evaluation is restricted to the ZINC dataset, which limits the assessment of generalization.

[1] Shi, Chence, et al. "Graphaf: a flow-based autoregressive model for molecular graph generation." arXiv preprint arXiv:2001.09382 (2020).

[2] Geng, Zijie, et al. "De novo molecular generation via connection-aware motif mining." arXiv preprint arXiv:2302.01129 (2023).

[3] Hetzel, Leon, et al. "MAGNet: Motif-Agnostic Generation of Molecules from Scaffolds." The Thirteenth International Conference on Learning Representations. 2025.

[4]  Kong, Xiangzhe, et al. "Molecule generation by principal subgraph mining and assembling." Advances in Neural Information Processing Systems 35 (2022): 2550-2563.

**Questions:**

- Could you include additional comparisons with the proposed baselines above for both unconditional and conditional generation tasks?

- How does the proposed method perform on scaffold-constrained generation and optimization tasks, following the experimental settings in JT-VAE and MoLeR?

---

> ### Author Response · Authors · 2025-11-17
> **Authors comment**
>
> Weakness 1.
>
> MolMiner shares the fragment-based autoregressive paradigm with MoLeR, but the use of a GMM serves different purposes. In MoLeR, the GMM approximates regions of the latent prior associated with specific scaffolds and is used as a sampling distribution for specific scaffold targeting. In contrast, our GMM operates directly in property space and is used to complete partially specified conditioning vectors, enabling users to provide any subset of the 12 supported molecular properties. The conditional distributions of the GMM allow us to sample the remaining entries in closed form, which is highly desirable for flexible multi-property inverse design. This capability is not present in MoLeR. While they use the same type of mixture model, the task is different.
>
> Weakness 2.
>
> In our setting, structural conditioning refers to steering the molecule’s composition and geometry through its physicochemical properties. Our focus in this work is broad property-driven inverse design, motif or scaffold-specific conditioning can be added in future extensions.
>
> Weakness 3.
>
> Our related work section focuses intentionally on autoregressive fragment-based models, which are the closest to our approach. One-shot models are noted in the introduction, but we restrict detailed discussion to the most comparable models.

---

### Official Review · Reviewer_nYie · 2025-10-27

**Soundness:** 2
**Presentation:** 2
**Contribution:** 2
**Rating:** 2
**Confidence:** 4

**Summary:**

MolMiner is a fragment-based autoregressive molecular generator that integrates dynamic geometry updates, symmetry-aware fragment attachment, and multi-property conditional control within a unified model. A geometry-informed attention mechanism and order-agnostic generation protocol aim to preserve molecular symmetry and physical plausibilitiy. For controllability, the authors propose a Gaussian Mixture Model (GMM)-based prior that allows conditioning on arbitrary subsets of twelve RDKit-computed molecular properties, with the remaining properties jointly sampled from the learned latent distribution. The paper further introduces calibtration-based evaluation metrics to assess conditional accuracy and distributional alignment.

**Strengths:**

1. The model can condition on any subset of 12 molecular properties, and demonstrates strong conditional generation results across all 12  properties.
2. The authors circumvent the lack of atomic matching and index tracking present in RDKit SMILES by maintaining atomic correspondences between individual building blocks and combined products.

**Weaknesses:**

- The geometric component is limited to a learned attention-bias term and does not involve explicit 3D coordinate inputs or conformer supervision. Consequently, the model remains fundamentally two-dimensional, and the paper’s central claim of geometry-conditioned control is slightly overstated.
- Across both unconditional and conditional benchmarks, MolMiner fails to outperform its primary comparator, HierVAE. Reported Wasserstein distances and calibration curves show only marginal or inferior performance, despite significantly higher architectural complexity. The results therefore do not substantiate the claimed advances in controllability or generative quality.
- MolMiner only compares molecular generation results to a single benchmark, HierVAE. Similar generative models (1, 2, 3) are not mentioned or benchmarked against.
- The properties listed do not include protein-conditioned binding affinity metrics, a critical use case for small molecule generation models.


1. Jin, W., Barzilay, R., & Jaakkola, T. (2019). Junction Tree Variational Autoencoder for Molecular Graph Generation. arXiv [Cs.LG]. Retrieved from http://arxiv.org/abs/1802.04364
2. Noutahi, E., Gabellini, C., Craig, M., Lim, J. S. C., & Tossou, P. (2023). Gotta be SAFE: A New Framework for Molecular Design. arXiv [Cs.LG]. Retrieved from http://arxiv.org/abs/2310.10773
3. Shi, C., Xu, M., Zhu, Z., Zhang, W., Zhang, M., & Tang, J. (2020). GraphAF: a Flow-based Autoregressive Model for Molecular Graph Generation. arXiv [Cs.LG]. Retrieved from http://arxiv.org/abs/2001.09382

**Questions:**

1. Is the random selection of open sites controllable for conditional generation problems? It seems rather important to select where to append fragments in experiments of that nature.
2.  It is mentioned that conditioning on more properties improves performance, but does this also yield lower diversity?
3. In section 3.4, the authors mention that geometry is "relaxed after each step via a classical force field", but this is not elaborated on elsewhere in the manuscript. Does MolMiner maintain an internal conformer or spatial atomic configuration?

---

> ### Author Response · Authors · 2025-11-17
> **Authors comment**
>
> Weakness 1:
>
> The importance of geometry in guiding the fragment selection is evaluated in Figure 6 section A.3.2 “The effect of geometry”, showing that the model does not behave the same with or without geometry, and demonstrating that geometry allows the model to make a better prediction of which token/fragment to attach next.
>
> MolMiner encodes geometry through the pairwise inter-fragment distance matrix, capturing the relative 3D arrangement of fragments while being invariant to global rotations and translations. Because every force-field relaxed conformer yields a different distance matrix, the model effectively receives 3D conformer supervision at each step.
>
> Weakness 2:
>
> A direct comparison to HierVAE on the 12-conditional benchmark is not possible because HierVAE is an unconditional model, and does not support multi-property inverse design. Our unconditional results therefore can not be evaluated against it.
>
> For the unconditional benchmark we report the differences in performance are small, and Fig. 15 provides a more detailed depiction of the distributional shifts. We acknowledge that specially for properties like molecular weight, TPSA and MR our model underperforms the predecessor. This arises from an early-termination bias introduced by the order-agnostic rollouts, which can lead to some generations ending prematurely and producing slightly smaller molecules, shifting these property distributions relative to the training data and thus the Wasserstein distances are greater (FIg.15).
>
> Weakness 3.
>
> Reference 1 (JTVAE) is discussed in the manuscript. Junction Tree Variational Autoencoder is the direct predecessor to HierVAE from the same research group, and HierVAE was introduced as an improved fragment-based model. For this reason, we compare against the latter, the strongest model in that lineage.
>
> Weakness 4.
>
> We do not include protein conditioned binding affinity metrics, but we do include 12 properties to do multi-objective inverse design. Binding affinity could be incorporated as an additional conditioning dimension in future applications. For this study, we intentionally selected 12 broadly applicable physico-chemical and structural properties that are relevant across diverse domains, including drug discovery, materials science, and energy applications like redox-flow batteries. To demonstrate general purpose multi-objective inverse design.
>
> Question 1.
>
> The random selection of open sites is a uniform sample from the queue of “site to explore”. This is intentionally by design, to allow for order-agnostic rollouts and maximum number of possible rollouts per molecule. The resulting molecule is subject only to the design constraints and thus invariant to the order at which its fragments are created. The generation process of a molecule has no canonical order, is order-invariant.
>
> Learning or enforcing a specific generative order would only be necessary for modeling synthetic pathways, which is outside the scope of this work and of most molecular generative models.
>
> Question 2.
>
> We show that conditioning on more properties improves performance in section A.3.1. The tomographic effect, Figure 5. We did not explicitly measure diversity under different conditioning subsets, but in general, specifying more target properties necessarily reduces the size of the feasible molecular set. As more constraints are imposed, the number of molecules that satisfy all of them decreases accordingly.
>
> Question 3.
>
> The conformer is reset at every docking step, ensuring that the geometry is not frozen but that is dynamically updated as the molecule grows. The conformation used to inform the next prediction step is the lowest energy conformation found in the relaxation.

---

### Official Review · Reviewer_zE68 · 2025-11-01

**Soundness:** 1
**Presentation:** 2
**Contribution:** 1
**Rating:** 0
**Confidence:** 5

**Summary:**

This paper propose an autoregressive model for molecular generation which is order-agnostic and enables conditional generation across multiple molecular properties. Notably, the method explicitly handles symmetry constraints and incorporates force-field–based geometric optimization during the generation process. However, the set of baseline comparisons is not comprehensive, the experimental results are relatively weak, and the overall technical contribution of the work remains limited.

**Strengths:**

- Order-agnostic autoregressive generation, enabling flexible and diverse molecular construction without fixed generation order.

- Multi-property conditional control, supporting simultaneous conditioning across multiple physicochemical and structural properties.

- Symmetry-aware fragment attachment and force-field–based geometry optimization, ensuring chemically consistent and physically realistic molecule generation.

**Weaknesses:**

- HierVAE was proposed several years ago and no longer represents the current state of the art in molecular generation. The paper lacks comparisons with more recent and stronger baselines, such as diffusion-based or flow-based models.

```
Song, Yuxuan, et al. "Equivariant flow matching with hybrid probability transport for 3d molecule generation." Advances in Neural Information Processing Systems 36 (2024).
Xu, Minkai, et al. "Geometric latent diffusion models for 3d molecule generation." International Conference on Machine Learning. PMLR, 2023.
Hoogeboom, Emiel, et al. "Equivariant diffusion for molecule generation in 3d." International conference on machine learning. PMLR, 2022.
Peng, Xingang, et al. "MolDiff: addressing the atom-bond inconsistency problem in 3D molecule diffusion generation." Proceedings of the 40th International Conference on Machine Learning. 2023.
Vignac, Clément, et al. "MiDi: Mixed Graph and 3D Denoising Diffusion for Molecule Generation." Joint European Conference on Machine Learning and Knowledge Discovery in Databases. 2023.
Dunn, Ian, and David Ryan Koes. "Mixed Continuous and Categorical Flow Matching for 3D De Novo Molecule Generation." ArXiv (2024).
```
- In Table 1, the molecular properties generated by HierVAE are closer to the reference data. The metrics of uniqueness, novelty, and diversity are not discriminative. Moreover, the task setting is relatively limited, the experimental comparisons are insufficient, and the paper lacks quantitative evaluation results.

**Questions:**

see weakness

---

> ### Author Response · Authors · 2025-11-17
> **Authors comment**
>
> We thank the reviewer for identifying strengths of the work, including the order-agnostic autoregressive generation, symmetry-aware fragment handling and dynamic geometry integration during generation. We now address the main concerns.
>
> 1. Baselines
>
> The referenced diffusion and flow models are strong approaches, but they operate on a different regime: 1. they generate atom-level 3D coordinates, often with fixed or narrowly varying atoms counts, 2. They do not enforce chemical validity by construction, and they do not let the model decide the size of the resulting molecule. 3. They do not support high dimensional conditioning (12 physicochemical properties) or evaluate calibration/alignment analysis on conditional generation across the full set of property ranges.
>
> MolMiner is a fragment-based autoregressive model, and appropriate baselines are other fragment-based autoregressive models. Among these, HierVAE is the most comparable architecture.
>
> 2. Unconditional benchmark performance
>
> We agree that HierVAE is slightly closer to the reference distributions, specially on Molecular weight, MR and TPSA, as a more detailed analysis in Fig. 15 illustrates further this distinction, putting it in perspective. However our model was designed and optimized for high-dimensional multi-property conditional inverse design. A capability absent in the baselines.
>
> 3. Contributions
>
> We disagree that the technical contribution is limited. To our knowledge, this is the first model to support:
> - order agnostic fragment-based autoregressive generation
> - subject to dynamic 3D geometry updates during generation
> - procedure to handle symmetries of fragments
> - scalable 12-dimensional conditional generation, with calibration curves across full ±2σ range for every property
>
> No prior fragment-based approach or 3D generative method unifies these capabilities in a single framework.
>
> 4. Quantitative evaluation
>
> Our work includes extensive quantitative results across 12 Wasserstein distances, uniqueness, novelty and diversity metrics (Tab.1), full distributional comparisons (Fig.15). Calibration plots for 12 properties (Fig.2), architecture and generation ablations (A.3, A.5, A.7, including Tab.3). Providing a thorough evaluation of both conditional and unconditional performance.
>
> In summary, MolMiner targets a different generative regime than diffusion/flow models: chemistry abiding, fragment-based controllable molecular design. Within this setting, we believe our contributions constitute a substantial and novel addition.

---

### Note · Authors · 2025-11-17

I have read and agree with the venue's withdrawal policy on behalf of myself and my co-authors.